# ENHANCING GENERATIVE AUTO-BIDDING WITH OFFLINE REWARD EVALUATION AND POLICY SEARCH

**Zhiyu Mou**[1*]  **Yiqin Lv**[1,2*†]  **Miao Xu**[1]  **Qi Wang**[2§]  **Yixiu Mao**[2]  **Jinghao Chen**[1,2†]
**Qichen Ye**[1]  **Chao Li**[1]  **Rongquan Bai**[1§]  **Chuan Yu**[1]  **Jian Xu**[1]  **Bo Zheng**[1]
[1]Taobao & Tmall Group of Alibaba, Beijing, China
[2]Department of Automation, Tsinghua University, Beijing, China
mouzhiyu.mzy@taobao.com, hhq123go@gmail.com, rongquan.br@taobao.com

## ABSTRACT

Auto-bidding is a critical tool for advertisers to improve advertising performance. Recent progress has demonstrated that AI-Generated Bidding (AIGB), which learns a conditional generative planner from offline data, achieves superior performance compared to typical offline reinforcement learning (RL)-based auto-bidding methods. However, existing AIGB methods still face a performance bottleneck due to their inherent inability to explore beyond the static dataset with feedback. To address this, we propose **AIGB-Pearl** (*Planning with EvaluAtor via **RL***), a novel method that integrates generative planning and policy optimization. The core of AIGB-Pearl lies in constructing a trajectory evaluator to assess the quality of generated scores and designing a provably sound KL-Lipschitz-constrained score-maximization scheme to ensure safe and efficient exploration beyond the offline dataset. A practical algorithm that incorporates the synchronous coupling technique is further developed to ensure the model regularity required by the proposed scheme. Extensive experiments on both simulated and real-world advertising systems demonstrate the state-of-the-art performance of our approach.

## 1 INTRODUCTION

The increasing demand for commercial digitalization has facilitated the development of the auto-bidding technique in online advertising. Distinguished from traditional manual bidding products, auto-bidding provides advertisers with an efficient and flexible scheme to automatically optimize bids in dynamic and competitive environments (Balseiro et al., 2021a; Deng et al., 2021; Balseiro et al., 2021b). Technically, auto-bidding constitutes an offline sequential decision-making problem that aims to maximize advertising performance over a bidding episode, constrained to a static offline dataset due to operational safety concerns (Mou et al., 2022).

As a standard approach to offline decision-making problems, offline reinforcement learning (RL) (Kumar et al., 2020; Mao et al., 2024b) is widely adopted to solve the auto-bidding problem. By employing conservative policy search schemes, offline RL mitigates the infamous *out-of-distribution* (OOD) problem (Fujimoto et al., 2019), enabling reliable generalization beyond the offline dataset. However, their reliance on bootstrapped value estimates renders offline RL methods prone to training instability (Peng et al., 2024), potentially compromising policy performance.

Recent advances in generative models shed new light on offline decision-making problems (Zhu et al., 2023; Kang et al., 2023). Specifically, AI-generated bidding (AIGB) models auto-bidding as a trajectory-generation task and employs a generative model to approximate the conditional trajectory distribution of the offline dataset (Guo et al., 2024). AIGB avoids bootstrapping and exhibits more stable training and superior performance. However, the modeling approach in AIGB does not explicitly align with the performance-maximization of the auto-bidding problem. Inherently, AIGB relies on imitating trajectories from the offline dataset (Ajay et al., 2023) and lacks the ability to improve its generation quality beyond the offline dataset based on the performance feedback. Consequently, its conditional generation in the extrapolation regime can become unreliable, potentially leading to performance degradation or even to risky trajectory generations.

---

*Equal contribution; †Work was done during an internship at Alibaba Group; §Corresponding authors.

Hence, there arises a question: *built on AIGB, the latest state-of-the-art auto-bidding method, can we devise a plausible scheme to involve policy optimization in its generative model?* To this end, a natural idea is to integrate offline RL methods into AIGB. However, it is nontrivial to implement in the auto-bidding problem since (i) there is a lack of reward signals in AIGB to guide the generative model. Specifically, the generation quality of the generative model remains unknown during training, making it infeasible to explore new trajectories beyond the offline dataset; (ii) no dedicated offline RL algorithm exists for AIGB. In particular, theoretical analysis that guarantees safe generalization and mitigates OOD issues for generative models in auto-bidding remains largely unexplored.

To address these critical challenges, we propose **AIGB-Pearl** (Planning with EvaluAtor via RL), an RL-enhanced version of AIGB that learns a *trajectory evaluator* to score generation quality and drive exploration of the generative model through continuous interaction. The evaluator is trained through supervised learning on the offline dataset. Crucially, to mitigate the OOD problem and ensure reliable use of the evaluator, we examine the theoretical upper bound on the evaluator's bias. Then, guided by this analysis, we formulate a *KL-Lipschitz-constrained* score-maximization objective with a provable suboptimality bound, enabling safe and effective exploration beyond the offline data. Moreover, to perform constrained score maximization, we design a practical algorithm that incorporates the *synchronous coupling* technique to satisfy the generative model's Lipschitz condition. In addition, we note that AIGB-Pearl does not require bootstrapping and exhibits greater training stability than offline RL methods.

To summarize, our contributions in this paper are fourfold: (i) we propose a novel generative auto-bidding method, AIGB-Pearl, that enables continuous improvement in generation quality through exploration beyond the offline dataset; (ii) we propose a provable KL-Lipschitz constrained score maximization objective with a sub-optimality bound, ensuring a safe and effective generalization beyond the offline dataset; (iii) we devise a practical algorithm with synchronous coupling that effectively ensures the Lipschitz requirement for the generative model; (iv) extensive simulated and real-world experiments demonstrate that AIGB-Pearl achieves SOTA performance and verify the effectiveness of the developed techniques in enhancing safe and effective generalization.

## 2 PRELIMINARIES

### 2.1 PROBLEM STATEMENT

This work studies the auto-bidding problem for a single advertiser subject to a budget constraint. The auction mechanism follows a sealed-bid, second-price rule. The objective is to devise a bidding policy that maximizes the cumulative value of the impressions won over a finite bidding episode (e.g., a day) within a budget $B > 0$. As established in (He et al., 2021), the optimal bid for each impression $i$ is proportional to its intrinsic value $v_i > 0$, scaled by a non-negative factor $a \geq 0$ that remains consistent across all impressions. Under this strategy, the advertiser wins an impression $i$ if $av_i \geq p_i$ and pays $p_i$ upon winning, where $p_i > 0$ is the market price. The Return on Investment (ROI) of impression $i$ is defined as $v_i/p_i$, and we denote its upper bound as $R_m \triangleq \max_i v_i/p_i$.

However, the scaling factor is unknown a priori, and impression volatility drives its continual change throughout the bidding process. Hence, a standard practice involves recalibrating the scaling factor $a$ at fixed intervals of $T \in \mathbb{N}_+$ time steps (Guo et al., 2024; He et al., 2021; Mou et al., 2022). This casts the auto-bidding to a sequential decision-making problem.

Specifically, the auto-bidding problem can be modeled as a Markov Decision Process (MDP) $< \mathcal{S}, \mathcal{A}, \mathcal{R}, \mathcal{P} >$. The state $s_t \triangleq [t, \bar{c}_{t-1}, x] \in \mathcal{S}$ is composed of the current time step $t \in [T]$, the cost ratio $\bar{c}_{t-1} = c_{t-1}/B > 0$ where $c_{t-1}$ is the advertiser's cost for impressions won between time step $t - 1$ and $t$, and a static advertiser-specific feature $x$ that includes the budget and many other individual information. The action $a_t \in \mathcal{A}$ denotes the calibrated scaling factor at time step $t$. The reward $r_t \geq 0$ describes the value of the impressions won between time steps $t$ and $t + 1$, and $\mathcal{P}$ denotes the state transition rule. The auto-bidding problem can be formulated as:

$$\max_{a_1, a_2, \cdots, a_T} \mathbb{E}_{s_{t+1} \sim \mathcal{P}(\cdot | s_t, a_t)} \left[ \sum_{t=1}^{T} r_t \right], \ \ \text{s.t.} \sum_{t=1}^{T} c_t \leq B. \tag{1}$$

**Offline Setting.** Due to safety concerns—common in real-world advertising systems—we are restricted to learning the optimal bidding policy from a static *offline dataset* $\mathcal{D}$ comprising historical

states and actions along with associated rewards. This makes the considered auto-bidding problem an offline sequential decision-making task.

## 2.2 OFFLINE RL METHODS

RL constitutes a standard approach for auto-bidding problems, seeking an optimal bidding policy $\pi : \mathcal{S} \to \mathcal{A}$ that maximizes cumulative reward. Specifically, this is typically achieved by learning a Q-value function, $Q(s_t, a_t) \triangleq \mathbb{E}_\pi[\sum_{t'=t}^{T} r_{t'}]$, through temporal difference (TD) error minimization:

$$\min_Q \ \mathbb{E}_{(s_t, a_t, r_t, s_{t+1}) \sim \mathcal{D}}[Q(s_t, a_t) - r_t - \max_{a_{t+1}} \hat{Q}(s_{t+1}, a_{t+1})]^2, \qquad (2)$$

where $\hat{Q}$ is a target Q-value function with parameters updated via Polyak averaging (Mnih et al., 2015). Upon convergence, the optimal bidding policy is derived as $\pi(s_t) = \arg\max_{a_t} Q(s_t, a_t)$.

Due to the offline setting of the considered auto-bidding problem, directly employing Eq. 2 results in the infamous *out-of-distribution* (OOD) problem (Fujimoto et al., 2019), making the policy erroneously deviate from the offline dataset $\mathcal{D}$ (Mao et al., 2024a). As a standard solution, offline RL (Yu et al., 2020; Kumar et al., 2020; Mao et al., 2023; Kidambi et al., 2020) constrains the policy's behavior near $\mathcal{D}$ during TD learning, enabling reliable generalization beyond the offline dataset.

However, offline RL methods are notoriously unstable due to training instability caused by TD-learning (Peng et al., 2024), in which the bootstrapped value of the Q-function serves as its training label, resulting in an erroneous ground truth. Training stability is critical in auto-bidding due to the absence of an accurate offline policy evaluation method and the high cost of online policy evaluation in real-world advertising systems (Mou et al., 2022).

## 2.3 GENERATIVE AUTO-BIDDING METHODS

**Definition 1** (Trajectory and Trajectory Quality). *The* trajectory *is formalized as the state sequence throughout the bidding episode, i.e.,* $\tau \triangleq [s_1, s_2, \cdots, s_T]$. *The trajectory quality is defined as the normalized cumulative reward of the trajectory, i.e.,* $y(\tau) \triangleq \sum_{t=1}^{T} \bar{r}_t$ [1], *where* $\bar{r}_t = r_t / B$.

Unlike RL methods, the AI-generated auto-bidding (AIGB) (Guo et al., 2024) treats the auto-bidding problem as a sequence generation task. Specifically, a conditional generative model is employed to fit the conditional trajectory distribution $p_\theta(\tau | y(\tau))$ within the offline dataset $\mathcal{D}$, i.e.,

$$\max_\theta \ \mathbb{E}_{(\tau, y(\tau)) \sim \mathcal{D}}[\log p_\theta(\tau | y(\tau))], \qquad (3)$$

where $\theta$ denotes the parameter. Let $y_m > 0$ be the maximum trajectory quality in $\mathcal{D}$, we have $\forall y \in \mathcal{D}, y \in [0, y_m]$. During inference, AIGB follows a *planning-and-control* architecture. Specifically, at each time step, a trajectory is sampled from the trained generative model that acts as the *planner*, with a manually set condition $y^* \triangleq (1 + \epsilon) y_m$, where $\epsilon > 0$ is a hyper-parameter. Then, an extra off-the-shelf inverse dynamic model (Agrawal et al., 2016), acting as the *controller*, is employed to compute the action. See Appendix B for detailed descriptions. AIGB avoids TD learning and generally outperforms offline RL methods (Guo et al., 2024).

However, AIGB relies on imitating trajectories from the offline dataset (Ajay et al., 2023) and lacks the ability to improve its generation quality beyond the offline dataset based on the performance feedback. Consequently, AIGB's conditional generation in the extrapolation regime ($y^* > y_m$) can be unreliable without explicit reward guidance, rendering exploration undirected and potentially leading to performance degradation or even risky trajectory generation. Furthermore, no theoretical guarantee exists for the quality of the generated trajectory in this extrapolation regime.

## 3 METHOD

Enabling AIGB to explore higher-quality trajectories beyond the offline dataset with explicit reward guidance can enhance its performance and generalization. To this end, we propose **AIGB-Pearl**

---

[1]Note that, as in real-world advertising systems, the bidding process will automatically suspend once the advertiser's budget runs out, and thereby, any action sequence will not violate the budget constraint. Hence, the trajectory quality can be directly defined as the cumulative reward of the trajectory.

(**P**lanning with **E**valu**A**tor via **RL**) that constructs a *trajectory evaluator* (hereinafter referred to as the *evaluator* for simplicity) to integrate RL methods into AIGB's planner training. Specifically, the evaluator learns a *score* $\hat{y}_\phi(\tau)$ to estimate the trajectory quality $y(\tau)$ via supervised learning based on the offline dataset $\mathcal{D}$, i.e., $\min_\phi \mathbb{E}_{\tau \sim \mathcal{D}}[(\hat{y}_\phi(\tau) - y(\tau))^2]$, where $\phi$ denotes the evaluator parameter. Then, with $\phi$ fixed, the planner tries to maximize the score of its generation through iterative interactions with the evaluator, as shown in Fig. 2. Formally, this can be formulated as:

$$\max_\theta \ L(\theta) \triangleq \mathbb{E}_{\tau \sim p_\theta(\tau|y^*)}[\hat{y}_\phi(\tau)], \tag{4}$$

where the condition is fixed to $y^*$ in both training and inference stages to ensure consistency.

As shown in Eq. 4, the effectiveness of AIGB-Pearl hinges critically on the evaluator's reliability. However, given the offline nature of the considered auto-bidding problem, the evaluator training is confined to the fixed dataset $\mathcal{D}$. Although we incorporate several techniques into the evaluator's supervised training to enhance its prediction accuracy (as detailed in Section 3.2.1 and Appendix E.1), directly applying Eq. 4 to the planner can still trigger the infamous OOD problem due to the evaluator's generalization limits outside the data support, potentially degrading the planner's true performance. This issue is particularly acute in auto-bidding, a risk-sensitive domain in which suboptimal or anomalous trajectory generation can result in substantial monetary losses or campaign failures. Notably, there remains a lack of theoretically principled approaches to this challenge.

To address this challenge, we first analyze the theoretical bounds on the evaluator's bias in Section 3.1. Then, guided by this analysis, we propose a KL-Lipschitz-constrained score-maximization objective for the planner to ensure reliable use of the evaluator. Notably, this objective is theoretically justified by a sub-optimality bound established in Section 3.1.1. Finally, a practical algorithm is presented in Section 3.2, which employs a synchronous coupling method to satisfy the planner's Lipschitz constraint in Section 3.2.2.

## 3.1 KL-LIPSCHITZ-CONSTRAINED SCORE MAXIMIZATION

This section focuses on the reliable exploitation of the evaluator-guided score maximization.

**Our basic idea** is to optimize $\theta$ within a domain where the gap between the planner's score $L(\theta)$ and its true performance $J(\theta) \triangleq \mathbb{E}_{\tau \sim p_\theta(\tau|y^*)}[y(\tau)]$ is bounded by a small certifiable upper bound. This ensures the score maximization occurs only in regions where the evaluator is reliable.

$$|J(\theta) - L(\theta)| = |\mathbb{E}_{\tau \sim p_\theta(\tau|y^*)}[y(\tau)] - \mathbb{E}_{\tau \sim p_\theta(\tau|y^*)}[\hat{y}_\phi(\tau)]|. \tag{5}$$

In the following, we investigate this gap. Specifically, we find that the trajectory quality $y(\tau)$ is Lipschitz continuous as stated in Theorem 1. The proof is given in Appendix C.1.

**Theorem 1** (Lipschitz Continuous of $y(\tau)$.)**.** *The trajectory quality $y(\tau)$ is $\sqrt{T}R_m$-Lipschitz continuous with respect to the Frobenius norm.*

Motivated by Theorem 1, we enforce a $\sqrt{T}R_m$-Lipschitz regularity on the evaluator's training to inherit the Lipschitz continuity of the true trajectory quality $y(\tau)$ (as described in Section 3.2.1). As the trained evaluator's Lipschitz constant may not equal $\sqrt{T}R_m$ exactly, we denote it as $k\sqrt{T}R_m$, where $k \geq 0$ quantifies the degree of violation. Note that a tighter adherence of the evaluator to the Lipschitz constraint yields a value of $k$ closer to 1.

Equipped with Theorem 1 and the Lipschitz property of the evaluator, we derive the following upper bound on the performance gap between $J(\theta)$ and $L(\theta)$, and the proof is given in Appendix C.2.

**Theorem 2** (Evaluator Bias in Planning Performance Bound)**.** *Let the upper bound of the evaluator's bias on its training dataset $\mathcal{D}$ be $\delta_D > 0$, i.e., $\mathbb{E}_{\tau \sim D}|y(\tau) - \hat{y}_\phi(\tau)| \leq \delta_D$, and let the Lipschitz constant of $\hat{y}_\phi(\tau)$ be $k\sqrt{T}R_m$. The gap between the planner's score $L(\theta)$ and its true performance $J(\theta)$ can be bounded by:*

$$|J(\theta) - L(\theta)| \leq \delta_D + (1+k)\sqrt{T}R_m \mathbb{E}_{y \sim p_D(y)} \left[ \underbrace{W_1(p_\theta(\tau|y^*), p_\theta(\tau|y))}_{\textit{Generation sensitivity to } y} + \underbrace{W_1(p_\theta(\tau|y), p_D(\tau|y))}_{\textit{Imitation error on } \mathcal{D}} \right],$$

*where $W_1$ denotes the 1-Wasserstein distance.*

Note that $\delta_D$ could be regulated to a small value via supervised training of the evaluator, and $k$ depends on the Lipschitz property of the resulting evaluator as stated before [2]. Consequently, bounding the evaluator bias in the planner's performance requires constraining the following two factors:

- the planner's generation sensitivity to condition $y$ (the first Wasserstein term)
- the planner's imitation error on the offline dataset (the second Wasserstein term).

Specifically, we establish that the first Wasserstein term can be bounded by the Lipschitz constant $\text{Lip}_{W_1}(p_\theta(\tau|y))$ of the planner with respect to the condition $y$ measured by $W_1$:

$$\mathbb{E}_{y \sim p_D(y)}[W_1(p_\theta(\tau|y^*), p_\theta(\tau|y))] \leq (1+\epsilon)y_m \text{Lip}_{W_1}(p_\theta(\tau|y)). \tag{6}$$

The proof is given in Appendix C.3. Therefore, to ensure the boundedness of the first Wasserstein term, we constrain the planner's Lipschitz constant $\text{Lip}_{W_1}(p_\theta(\tau|y))$ to a positive hyperparameter $L_p$. A lower bound analysis of $L_p$ is provided later in Eq. 10.

Moreover, we establish that a constrained KL divergence $\mathbb{E}_{y \sim p_D(y)}[D_{\text{KL}}(p_D(\tau|y)\|p_\theta(\tau|y))] \leq \delta_K$ could bound the expectation of the second Wasserstein distance term as follows, where $\delta_K > 0$ is a hyperparameter and can be set to a small value, close to zero. See Appendix C.4 for the proof. Note that the KL divergence constraint here inherently makes the planner perform conditional behavior cloning on the offline dataset $\mathcal{D}$ (Guo et al., 2025).

$$\mathbb{E}_{y \sim p_D(y)}[W_1(p_\theta(\tau|y), p_D(\tau|y))] \leq \sqrt{\delta_K}. \tag{7}$$

Collectively, to effectively perform score maximization with a small, certifiable evaluator bias, we enforce Lipschitz continuity of the planner with respect to the condition $y$, while preserving its behavior cloning fidelity to the offline dataset $\mathcal{D}$. Formally, Eq. 4 is transformed to:

$$\max_\theta \quad L(\theta) \quad \text{(Score Maximization)} \tag{8}$$

$$\text{s.t.} \quad \mathbb{E}_{y \sim p_D(y)}[D_{\text{KL}}(p_D(\tau|y)\|p_\theta(\tau|y))] \leq \delta_K \quad \text{(KL Constraint)} \tag{8a}$$

$$\text{Lip}_{W_1}(p_\theta(\tau|y)) \leq L_p \quad \text{(Lipschitz Constraint)} \tag{8b}$$

Eq. 8 forms the score maximization objective in AIGB-Pearl.

**Remark 1.** *Intuitively, the KL and Lipschitz constraints jointly ensure the planner's generation under condition $y^*$ remains within a certified neighborhood of the high-quality trajectories in the offline dataset $\mathcal{D}$. As illustrated in Fig. 1, the green region represents the feasible set of trajectories $p_\theta(\tau|y)$ satisfying the KL constraint for $y \in \mathcal{D}$ [3]. The Lipschitz constraint makes the generated trajectories $p_\theta(\tau|y^*)$ remain within a neighborhood of the best-quality trajectory in the offline dataset $p_\theta(\tau|y_m)$, as illustrated by the blue circle with radius $\epsilon L_p y_m$. The union of all such circles constitutes the total trajectory exploration range. Meanwhile, the evaluator trained on $\mathcal{D}$ maintains high accuracy within this $\mathcal{D}$-proximal region, and the Lipschitz regulariza-*

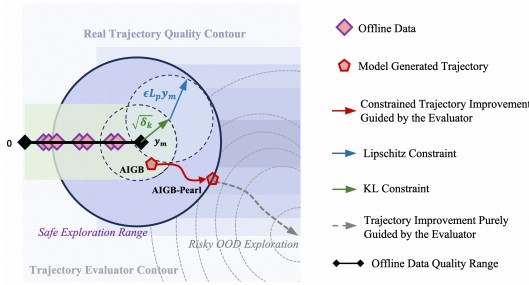

Figure 1: Schematic illustration of the KL-Lipschitz-constrained score-maximization. It enables safe trajectory improvement within a theoretically certified neighborhood of the high quality trajectories in the offline dataset, where the evaluator remains high accuracy.

*tion on the evaluator bounds its sensitivity to input perturbations, preventing drastic value fluctuations in OOD regions and promoting more reliable extrapolation. Overall, KL-Lipschitz constrained optimization, guided by a Lipschitz continuity evaluator, enables safe trajectory improvement within a theoretically certified neighborhood of high-quality offline trajectories, effectively mitigating risky OOD exploration.*

---

[2] Note that $k$ cannot approach zero without compromising $\delta_D$, as excessively small $k$ prevents the evaluator from fitting the offline dataset $\mathcal{D}$.

[3] Note that the KL-constraint feasible region may not be a regular topology since the generated trajectories $p_\theta(\tau|y)$ should also satisfy the Lipschitz constraint under condition $y \in \mathcal{D}$.

### 3.1.1 SUB-OPTIMALITY GAP BOUND

This section focuses on presenting and analyzing the sub-optimality bound of the solution to the proposed Eq. 8. Specifically, denote the solution to the true performance $J(\theta)$ as $\theta^* \triangleq \arg\max_\theta J(\theta)$, and denote the solution to the proposed Eq. 8 as $\hat{\theta}$. The following theorem gives the sub-optimality bound of the planner's performance, and the proof is given in Appendix C.5.

**Theorem 3** (Sub-optimality Gap Bound). *Let $\delta_M \triangleq \mathbb{E}_{y \sim p_D(y)}[D_{KL}(p_D(\tau|y)\|p_{\theta^*}(\tau|y^*))]$ be the expected distance between the optimal trajectory distribution and the trajectory distribution of the offline dataset $\mathcal{D}$. The true performance gap between the optimal parameter $\theta^*$ and the solution $\hat{\theta}$ to Eq. 8 is bounded by:*

$$J(\theta^*) - J(\hat{\theta}) \leq 2\delta_D + (1+2k)\sqrt{T}R_m\left[\sqrt{\delta_M} + \sqrt{\delta_K} + (1+\epsilon)y_m L_p\right]. \tag{9}$$

**Theoretical Result Analysis.** In Theorem 3, the constants $\delta_M, R_m, T, \epsilon, y_m$ characterize domain-specific properties of the auto-bidding task and the offline dataset $\mathcal{D}$. Nonetheless, a lower training error $\delta_D$ of the evaluator and a closer $k$ to 1 correspond to a smaller sub-optimality gap. Note that $k$ cannot be smaller than 1 without compromising $\delta_D$, as excessively small $k$ prevents the evaluator from fitting the offline dataset $\mathcal{D}$.

Moreover, in Theorem 3, a lower behavior cloning error $\delta_K$ and a lower Lipschitz constant $L_p$ of the planner lead to a smaller sub-optimality gap. However, an excessively small $L_p$ prevents the planner from behavior cloning the offline dataset $\mathcal{D}$ (as required by the KL constraint), resulting in a large $\delta_K$. Actually, a theoretical lower bound for $L_p$ is given by the Lipschitz constant of the conditional trajectory distribution of the offline dataset $p_D(\tau|y)$:

$$L_p \geq \sup_{y_1 \neq y_2} \frac{W_1(p_D(\tau|y_1), p_D(\tau|y_2))}{|y_1 - y_2|}. \tag{10}$$

where $y_1, y_2 \in \mathcal{D}$. Consequently, we leverage this lower bound of $L_p$ in AIGB-Pearl. A further discussion on the theoretical performance range of AIGB-Pearl is given in Appendix D.

## 3.2 PRACTICAL ALGORITHM DESIGN

This section focuses on the practical algorithm implementation of Eq. 8. Section 3.2.1 first presents our reliability-enhanced evaluator architecture, followed by the synchronous-coupling-based Lipschitz planner design in Section 3.2.2.

### 3.2.1 LIPSCHITZ TRAJECTORY EVALUATOR

As shown in Fig. 2, the evaluator processes the trajectory $\tau$ to predict a score $\hat{y}_\phi(\tau)$ for quality estimation. The evaluator is trained via supervised learning using the offline dataset $\mathcal{D}$. Besides, to satisfy $\sqrt{T}R_m$-Lipschitz constraint requirement according to Theorem 2, we add Lipschitz regularization term to the training loss of the evaluator, which can be expressed as:

$$l_e(\phi) = \underbrace{\mathbb{E}_{\tau \sim \mathcal{D}}\left[(\hat{y}_\phi(\tau) - y(\tau))^2\right]}_{\text{fitting the ground truth}} + \beta_1 \underbrace{\mathbb{E}_{\tau_1, \tau_2}\left[|\hat{y}_\phi(\tau_1) - \hat{y}_\phi(\tau_2)| - \sqrt{T}R_m\|\tau_1 - \tau_2\|_F\right]_+}_{\text{Lipschitz penalty}}, \tag{11}$$

where $\beta_1 > 0$ is a hyper-parameter, $[\cdot]_+ \triangleq \max\{0, \cdot\}$. Moreover, to further enhance the prediction accuracy of the evaluator, we devise two specific techniques, including the LLM Embedding enhancement and pair-wise learning, whose details are given in Appendix E.1.

### 3.2.2 LIPSCHITZ PLANNER WITH SYNCHRONOUS COUPLING

As shown in Fig. 2, the planner is implemented by a Causal Transformer (Chen et al., 2021) that generates trajectories in an auto-regressive manner. Specifically, the model takes the condition $y$ and history states $s_{1:t}$ as input tokens, and predicts the next state as a Gaussian distribution, $p_\theta(s_{t+1}|s_{1:t}, y) = \mathcal{N}(\mu_\theta(s_{1:t}, y, t), \sigma_\theta^2(s_{1:t}, y, t))$, where $\mu_\theta$ denotes the mean and $\sigma_\theta > 0$ the standard deviation. During the auto-regressive generation process, each output state is sampled from the

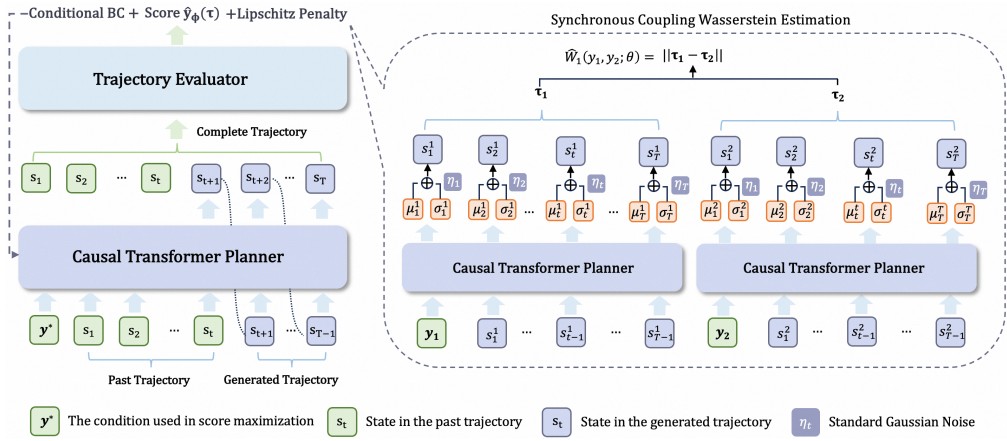

Figure 2: **AIGB-Pearl** *(Planning with EvaluAtor via RL)* constructs a trajectory evaluator to score the trajectory quality and let the planner maximize the obtained score under the KL-Lipschitz constraint through continuous interaction with the evaluator. A synchronous coupling method is used to estimate the Wasserstein term in the Lipschitz penalty.

Gaussian distribution using the reparameterization trick, i.e., $s_{t+1} = \mu_\theta(s_{1:t}, y, t) + \sigma_\theta(s_{1:t}, y, t) \cdot \eta_t$, where $\eta_t \sim \mathcal{N}(0, I)$ [4].

**Regularized Planner Training Loss.** To perform the score maximization in Eq. 8, we involve two regularization terms in the planner's training loss $l_p(\theta)$, including a conditional behavior cloning loss, corresponding to the KL constraint Eq. 8a, and a Lipschitz penalty loss, corresponding to the Lipschitz constraint Eq. 8b, i.e.,

$$l_p(\theta) = - \underbrace{\mathbb{E}_{\tau \sim p_\theta(\tau|y^*)}[\hat{y}_\phi(\tau)]}_{\text{planner score } L(\theta)} - \beta_2 \underbrace{\mathbb{E}_{(\tau,y) \sim p_D}[\log p_\theta(\tau|y)]}_{\text{conditional behavior clone}}$$

$$+ \beta_3 \underbrace{\mathbb{E}_{y_1, y_2 \in \mathcal{D} \cup \{y^*\}} \left[ W_1(p_\theta(\tau|y_1), p_\theta(\tau|y_2)) - L_p|y_1 - y_2| \right]_+}_{\text{Lipschitz penalty, where } W_1(p_\theta(\tau|y_1), p_\theta(\tau|y_2)) \text{ is replaced by } \hat{W}_1(y_1, y_2; \theta)}, \quad (12)$$

where $\beta_2, \beta_3 > 0$ are two hyper-parameters. With prior RL works (Sutton et al., 1999), we can derive the closed-form expression of planner's score gradient $\nabla_\theta L(\theta)$ as shown in Appendix C.6. The core of the planner loss lies in the computation of $W_1(p_\theta(\tau|y_1), p_\theta(\tau|y_2))$.

**Wasserstein Upper Bound as Surrogate.** Accurate computation of this Wasserstein distance term is challenging, as it requires finding the optimal coupling between $p_\theta(\tau|y_1)$ and $p_\theta(\tau|y_2)$ that minimizes the expected transportation cost. Nonetheless, we can choose a certain coupling $\gamma \in \Gamma(p_\theta(\tau|y_1), p_\theta(\tau|y_2))$ to obtain an upper bound of this Wasserstein term, i.e.,

$$W_1(p_\theta(\tau|y_1), p_\theta(\tau|y_2)) \triangleq \inf_{\gamma \in \Gamma(p_\theta(\tau|y_1), p_\theta(\tau|y_2))} \mathbb{E}_{(\tau_1, \tau_2) \sim \gamma} \left[ \sum_t \|s_t^1 - s_t^2\| \right]$$

$$\leq \mathbb{E}_{\eta_{1:T} \sim \mathcal{N}(0, I)} \left[ \sum_t \|s_t^1 - s_t^2\| \right] \triangleq \hat{W}_1(y_1, y_2; \theta). \quad (13)$$

where $\hat{W}_1(y_1, y_2; \theta)$ denotes the upper bound, and $s_t^i$ is the $t$-th state in trajectory $\tau_i$. It can be seen that $\hat{W}_1(y_1, y_2; \theta) \leq L_p|y_1 - y_2|$ acts as a sufficient condition to make the planner $L_p$-Lipschitz continuous. Thus, we replace $W_1(p_\theta(\tau|y_1), p_\theta(\tau|y_2))$ by this upper bound in the planner loss.

**Synchronous Coupling Wasserstein.** Instead of using random couplings, we employ a *synchronous coupling* $\gamma_{\text{sync}}$ to make the upper bound tighter. Specifically, two trajectories $\tau_1$ and $\tau_2$—conditioned on $y_1$ and $y_2$, respectively—are generated using the same sequence of Gaussian noise $\{\eta_1, \eta_2, ... \eta_T\}$.

---

[4]Note that the time step $t$ and the static advertiser feature $x$ in the state $s_t = [t, c_{t-1}, x]$ do not need to be generated. We only generate the next cost ratio $c_t$ in practice.

Table 1: Overall performance (GMV) in simulated experiments with 30 advertisers. $\Delta$ indicates the relative improvement of AIGB-Pearl against the most competitive baseline (which is underlined). Note that the absolute values are normalized without specific meanings; only $\Delta$ matters.

| Budget | USCB | BCQ | CQL | IQL | MORL | MOPO | DT | DiffBid | AIGB-Pearl | $\Delta$ |
|---|---|---|---|---|---|---|---|---|---|---|
| 1.5k | 454.25 | 454.72 | 461.82 | 456.80 | 468.49 | 470.38 | 477.39 | 480.76 | 502.98 | **+4.62%** |
| 2.0k | 482.67 | 483.50 | 475.78 | 486.56 | 488.12 | 489.27 | 507.30 | 511.17 | 521.84 | **+2.09%** |
| 2.5k | 497.66 | 498.77 | 481.37 | 518.27 | 511.93 | 523.91 | 527.88 | 531.29 | 545.03 | **+2.59%** |
| 3.0k | 500.60 | 501.86 | 491.36 | 549.19 | 553.91 | 549.01 | 550.66 | 556.32 | 574.17 | **+3.21%** |

Table 2: Overall performance in real-world A/B tests, involving 6k advertisers over 19 days.

| Methods | GMV | BuyCnt | ROI | Cost | Methods | GMV | BuyCnt | ROI | Cost |
|---|---|---|---|---|---|---|---|---|---|
| **DiffBid** | 76,390,174 | 650,962 | 5.31 | 14,395,290 | **USCB** | 52,182,805 | 516,994 | 4.92 | 10,598,486 |
| **AIGB-Pearl** | 78,676,009 | 665,173 | 5.41 | 14,551,054 | **AIGB-Pearl** | 53,973,101 | 520,796 | 5.13 | 10,515,772 |
| $\Delta$ | **+3.00%** | **+2.20%** | **+1.89%** | +1.10% | $\Delta$ | **+3.43%** | **+0.74%** | **+4.24%** | -0.78% |

| Methods | GMV | BuyCnt | ROI | Cost | Methods | GMV | BuyCnt | ROI | Cost |
|---|---|---|---|---|---|---|---|---|---|
| **DT** | 34,808,665 | 341,995 | 5.61 | 6,205,665 | **MOPO** | 51,674,071 | 579,332 | 3.08 | 16,771,892 |
| **AIGB-Pearl** | 35,957,933 | 344,194 | 5.77 | 6,246,512 | **AIGB-Pearl** | 53,292,945 | 591,741 | 3.23 | 16,475,670 |
| $\Delta$ | **+3.30%** | **+0.64%** | **+0.16%** | +0.66% | $\Delta$ | **+3.13%** | **+2.14%** | **+4.87%** | -1.77% |

The definition of $\hat{W}_1(y_1, y_2; \theta)$ is given in Eq. 13. Compared to random couplings, the synchronous coupling reduces spurious variance in the trajectory comparison by aligning stochasticity through shared noise, resulting in a tighter upper bound on the Wasserstein distance (Lindvall, 2002).

Moreover, if we make the predicted variance $\sigma_\theta$ of the planner a fixed constant, then the expression of $\hat{W}_1(y_1, y_2; \theta)$ can be further simplified to $\hat{W}_1(y_1, y_2; \theta) = \sum_t \|\mu_\theta(s^1_{1:t}, y_1, t) - \mu_\theta(s^2_{1:t}, y_2, t)\|$. The overall AIGB-Pearl algorithm is summarized in Algorithm 1 in Appendix E due to page limits.

## 4 EXPERIMENTS

We conduct both simulated and real-world experiments to validate the effectiveness of our approach. In the experiments, we mainly investigate the following Research Questions (**RQ**s): (1) Does enhancing AIGB with policy optimization improve overall performance, and can it generalize better to unseen data compared to existing AIGB methods? (Section 4.2) (2) How does the KL-Lipschitz constraint affect the performance of the planner? (Section 4.3) (3) Can the proposed method guarantee the Lipschitz property of the evaluator and the planner? (Section 4.4). (4) What is the evaluator's accuracy on the training data, and how well does it generalize to unseen data? (Section 4.5). The training stability of AIGB-Pearl is studied in Appendix F.5.

### 4.1 EXPERIMENT SETUP

**Experiment Environment.** We conduct simulated experiments in an open-source offline advertising system with 30 advertisers of four budget levels (1.5k, 2.0k, 2.5k, and 3.0k), as in (Mou et al., 2022; Guo et al., 2024). The offline dataset comprises 5k trajectories generated by 20 advertisers. Extra detailed settings of simulated experiments are given in Appendix F.1. For real-world experiments, we conduct online A/B tests on one of the world's largest E-commerce platforms, TaoBao. The offline dataset comprises 200k trajectories of 10k advertisers. See Appendix F.2 for extra detailed settings of real-world experiments. In both simulated and real-world experiments, we employ the same inverse dynamics model from Agrawal et al. (2016) as the controller in AIGB. Moreover, the evaluator is trained on the entire offline dataset, and its generalization ability is evaluated using $K$-fold cross-validation with $K = 5$.

**Baselines.** We compare our method with state-of-the-art AIGB methods, including **DiffBid** (Guo et al., 2024) and **DT** (Chen et al., 2021), which learn from conditional behavior cloning of offline datasets using a diffusion model and a Causal Transformer, respectively. We also compare our method with RL auto-bidding methods, including **USCB** (He et al., 2021) that learns the auto-bidding policy in a manually constructed advertising system with DDPG (Silver et al., 2014); and offline RL auto-bidding methods, including model-free offline RL methods **BCQ** (Fujimoto et al., 2019), **CQL** (Kumar et al., 2020), and **IQL** (Kostrikov et al., 2022), and model-based offline RL methods **MOPO** (Yu et al., 2020) and **MORL** (Mou et al., 2025).

Table 3: Generalization performance in real-world A/B tests with unseen advertisers against AIGB methods, involving 4k advertisers over 19 days.

| Methods | GMV | BuyCnt | ROI | Cost | Methods | GMV | BuyCnt | ROI | Cost |
|---|---|---|---|---|---|---|---|---|---|
| **DiffBid** | 67,092,973 | 553,020 | 5.39 | 12,444,306 | **DT** | 30,562,007 | 300,271 | 5.61 | 5,450,573 |
| **AIGB-Pearl** | 69,252,539 | 565,776 | 5.53 | 12,534,379 | **AIGB-Pearl** | 31,502,309 | 305,202 | 5.74 | 5,484,473 |
| **Δ** | **+3.32%** | **+2.31%** | **+2.48%** | +0.72% | **Δ** | **+3.08%** | **+1.64%** | **+2.32%** | +0.62% |

Table 4: Ablation Study. The effectiveness of the KL constraint and the Lipschitz constraint in Real-world A/B tests, involving 6k advertisers over 8 days.

| AIGB-Pearl | GMV | BuyCnt | ROI | Cost | Methods | GMV | BuyCnt | ROI | Cost |
|---|---|---|---|---|---|---|---|---|---|
| **w/o KL** | 30,906,963 | 292,605 | 4.25 | 7,269,018 | **w/o Lipschitz** | 32,284,972 | 268,551 | 5.73 | 5,634,304 |
| **with KL** | 31,243,688 | 292,783 | 4.26 | 7,342,485 | **with Lipschitz** | 32,869,329 | 281,979 | 5.79 | 5,678,252 |
| **Δ** | **+1.09%** | **+0.06%** | **+0.08%** | +1.01% | **Δ** | **+1.81%** | **+0.50%** | **+1.05%** | +0.78% |

**Performance Index.** The objective in the auto-bidding problem Eq. 1, i.e., the cumulative rewards over the bidding episode, acts as the main performance index in our experiments and is referred to as the *gross merchandise volume*, **GMV**. In addition, we utilize three other metrics commonly used in the auto-bidding problem to evaluate the performance of our approach. The first metric is the total number of impressions won over the bidding episode, referred to as the **BuyCnt**. The second metric is the **Cost** over the bidding episode, and the third is the *return on investment* **ROI**, defined as the ratio of GMV to Cost. Note that larger values of GMV, BuyCnt, and ROI with a Cost oscillating within an acceptable tolerance ($\pm 2\%$) indicate a better performance.

## 4.2 OVERALL PERFORMANCE

**To answer RQ(1):** Table 1 shows that our method consistently outperforms all baselines in GMV across all four budget levels in simulated experiments. In real-world experiments, Table 2 shows that our method also achieves superior performance on GMV, BuyCnt, and ROI, with Cost fluctuations of less than 2%. Notably, both simulated and real-world experiments consistently demonstrate that AIGB-Pearl achieves a **+3%** improvement in GMV over the AIGB, the state-of-the-art auto-bidding method. Since our method and DiffBid share the same controller, the performance gain stems solely from the planner. This provides strong empirical evidence that the proposed conservative RL learning for score maximization effectively enhances overall performance.

Notably, we also apply AIGB-Pearl to another important auto-bidding problem, named TargetROAS (See Appendix F.3). Real-world experiments show that AIGB-Pearl achieves a **+5%** improvement in GMV compared to AIGB. It is worth noting that a GMV uplift exceeding 2% is highly significant, translating to **millions** of RMB in additional **daily** GMV on Taobao-scale advertising platforms.

**Generalization Ability.** We evaluate AIGB-Pearl on advertisers not used to generate trajectories in the offline dataset and compare it with existing AIGB methods. For simplicity, we refer to these advertisers as *advertisers outside the offline dataset*. Table 3 reports the performance on 4k advertisers outside the offline dataset in real-world experiments. AIGB-Pearl consistently delivers better results in GMV (**+3%**), BuyCnt, and ROI, while maintaining Cost fluctuations within 2% compared to the baselines. This indicates that the proposed method has better generalization ability than AIGB.

## 4.3 ABLATION STUDY

**To answer RQ(2):** We remove the KL and Lipschitz constraints from AIGB-Pearl individually and evaluate the model's performance in each ablated variant using real-world A/B tests. The results are presented in Table 4. It can be seen that the KL constraint contributes **+1.1%** improvement in GMV, and the Lipschitz constraint provides **+1.8%** improvement in GMV, demonstrating their respective roles in enhancing AIGB-Pearl's performance.

**Visualization.** Three AIGB-Pearl-generated trajectory examples are presented in Fig. 3. As can be observed, the trajectories generated by AIGB-Pearl are plausible. In contrast, the ablated variant without the KL and Lipschitz constraints produces trajectories that deviate significantly from the optimal trajectory in the offline dataset and exhibit clear pathological behaviors—such as excessive budget consumption, backward-trending pacing, and under-utilization of available budgets (see Appendix F.4 for explanation)—which further validate the KL-Lipschitz constraint necessity.

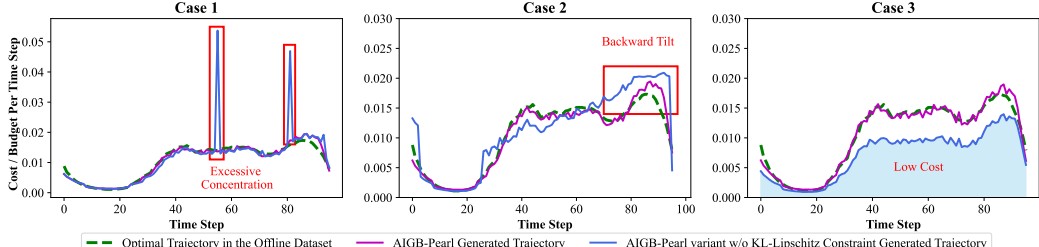

Figure 3: Trajectory Generation Visualization. Three cases are presented. Here, the AIGB-Pearl generates plausible trajectories, whereas its variant without the KL-Lipschitz constraint produces generations that significantly deviate from the reference and exhibit evident issues.

Table 5: Evaluator accuracy for simulated and real-world experiments. Results are reported for training data and OOD data evaluated using 5-fold cross-validation.

| Simulated Exp | Training Data | OOD Data (Cross-Validation) | Real-world Exp | Training Data | OOD Data (Cross-Validation) |
|---|---|---|---|---|---|
| MAE $\downarrow$ | 0.6 | $0.7 \pm 0.06$ | MAE $\downarrow$ | 1.0 | $1.2 \pm 0.03$ |
| AUC $\uparrow$ | 89.9% | $85.5\% \pm 0.5\%$ | AUC $\uparrow$ | 77.4% | $75.1\% \pm 0.2\%$ |

## 4.4 LIPSCHITZ VALUE EXAMINATION

**To answer RQ(3):** We report that the Lipschitz value of the trajectory quality $y(\tau)$ and the conditional trajectory distribution $p_D$ of the offline dataset are $1.62$ and $0.38$, respectively. We set $L_p = 0.50$, which is close to its lower bound estimate of $0.38$ [5]. To calculate the Lipschitz constants of the evaluator and the planner, we sample $8,000$ pairs of trajectories and compute their Lipschitz constants. The results are shown

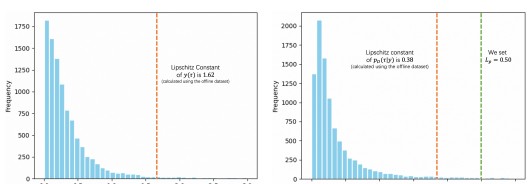

Figure 4: Examination of Evaluator Lipschitz.

Figure 5: Examination of Planner Lipschitz.

in Fig. 4 and Fig. 5. It can be observed that most sample values satisfy the Lipschitz constraint, and the Lipschitz constants of the evaluator $\hat{y}_\phi(\tau)$ and planner $p_\theta(\tau|y)$ are $2.2$ and $0.56$, respectively, near $1.62$ and $0.50$. This indicates that the models' Lipschitz constraints are successfully satisfied.

## 4.5 EVALUATOR ACCURACY EXAMINATION

**Accuracy Metrics.** The evaluator's accuracy is assessed along two dimensions, including the *absolute accuracy* measured by mean absolute error (MAE) metrics, reflecting how close the predicted scores are to ground truth scores, and the *ranking accuracy* by AUC metric, reflecting the correctness of relative rankings between trajectory pairs. Note that the MAE of each advertiser's data sample is normalized by its budget to ensure comparability across advertisers. A lower MAE, together with a higher AUC, indicates better evaluator accuracy.

**To answer RQ(4):** We report the accuracy of the trained evaluator in both simulated and real-world experiments in Table 5. We evaluate the evaluator's accuracy on the training data and its generalization ability using $K$-fold cross-validation, where $K = 5$. To the best of our knowledge, we are the first to introduce the trajectory evaluator into the generative auto-bidding framework. The reasonableness of our evaluator is evidenced by its pairwise ranking accuracies of 86% AUC and 75% AUC on OOD trajectories in simulated and real-world experiments, respectively, substantially above the 50% random-chance level, despite the high complexity and dynamic nature of the bidding environment. Importantly, with the guidance of the trained evaluator, the planner outperforms state-of-the-art AIGB methods even on OOD data, as demonstrated in the Table. 3.

## 5 CONCLUSIONS

This paper proposes AIGB-Pearl, which enhances AIGB by incorporating reward evaluation and policy optimization. By introducing a trajectory evaluator and a provably KL-Lipschitz-constrained score-maximization objective, our approach ensures safe and efficient generalization beyond the offline dataset, supported by theoretical guarantees. Extensive simulated and real-world experiments validate the state-of-the-art performance of our approach.

---

[5]As the data-driven lower bound estimation can be underestimated, we slightly increase its value in practice.

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

CONTENTS

## A    RELATED WORKS

### A.1    RL-BASED AUTO-BIDDING METHODS

Auto-bidding plays a critical role in online advertising by automatically placing bids, allowing advertisers to participate efficiently in real-time auctions (Balseiro et al., 2021a; Deng et al., 2021; Balseiro et al., 2021b). The auto-bidding problem can be modeled as a Markov Decision Process and addressed using reinforcement learning techniques. USCB (He et al., 2021) proposes a unified solution to the constrained bidding problem, employing the RL method DDPG (Silver et al., 2014) to dynamically adjust parameters toward an optimal bidding strategy. Mou et al. (2022) propose a sustainable online reinforcement learning framework that alternates between online exploration and offline training, thereby alleviating the sim2rel problem. A few studies explore multi-agent RL for auto-bidding (Jin et al., 2018; Guan et al., 2021; Wen et al., 2022), while several focus on budget allocation and bidding strategies in multi-channel scenarios using RL-based approaches (Wang et al., 2023; Deng et al., 2023; Duan et al., 2025). Importantly, offline RL methods such as BCQ (Fujimoto et al., 2019), CQL (Kumar et al., 2020), IQL (Kostrikov et al., 2022), and MOPO (Yu et al., 2020) have demonstrated significant potential in this domain. These methods allow policy learning from pre-collected datasets without requiring online interaction. Moreover, offline RL, such as Diffusion-QL (Wang et al., 2022b), employs a generative policy architecture to improve expressive capacity.

However, RL-based methods often suffer from training instability arising from bootstrapping and alternating training between critics and actors. Training instability typically deteriorates policy performance (Sutton et al., 1998). Moreover, training stability is even more critical in auto-bidding, given two domain-specific challenges: the absence of an accurate offline policy evaluation method and the high cost of online policy evaluation in a real-world advertising system (Mou et al., 2022). Therefore, stable convergence to a well-performed policy is essential to ensure deployment reliability and system safety.

### A.2    GENERATIVE AUTO-BIDDING METHODS

Generative models exhibit strong capabilities for capturing and reproducing underlying data distributions across a wide range of fields (Kingma & Welling, 2022; Goodfellow et al., 2020; Pan et al., 2023; Sohl-Dickstein et al., 2015; Ho et al., 2020; Vaswani et al., 2017). They can be effectively incorporated into decision-making systems by generating complete trajectories that guide agents toward high-reward behaviors (Zhu et al., 2023; Kang et al., 2023; Li et al., 2025). In particular, Decision Transformer (DT) (Chen et al., 2021) reframes RL as a conditional sequence modeling problem and leverages transformer architectures to generate actions conditioned on desired returns, historical states, and actions. AIGB (Guo et al., 2024) extends the generative perspective to the auto-bidding domain by formulating auto-bidding as a conditional generative modeling problem. DiffBid generates a state trajectory based on the desired return utilizing a conditional diffusion model, and then generates actions aligned with the optimized trajectory. These methods achieve superior performance in auto-bidding and offer distinct advantages over traditional RL methods. They do not rely on the bootstrapping mechanism commonly used in RL, thereby avoiding the instability caused by the deadly triad. Even so, these generative auto-bidding methods still face a performance bottleneck due to their neglect of fine-grained generation-quality evaluation and their inability to explore beyond static datasets. In contrast, our method facilitates both reward evaluation and policy search through a learned trajectory evaluator.

## B    AIGB METHOD DETAILS

AIGB models the sequential decision-making problem via conditional diffusion, enabling effective trajectory generation for auto-bidding scenarios. Specifically, AIGB utilizes the denoising diffusion probabilistic model (DDPM) (Ho et al., 2020) for generation. The forward and reverse processes are modeled as:

$$q(\tau_{k+1}|\tau_k), \quad p_\theta(\tau_k|\tau_{k+1}, y(\tau)), \tag{14}$$

respectively, where $q$ represents the forward noising process while $p_\theta$ the reverse denoising process.

**Forward Process.** In the forward process, the noise is gradually added to the latent variable by a Markov chain with pre-defined variance schedule $\beta_k$:

$$q(\tau_k|\tau_{k-1}) = \mathcal{N}(\tau_k; \sqrt{1 - \beta_k}\tau_{k-1}, \beta_k I) \tag{15}$$

where $k \in [K]$ refers to the diffusion step, $\tau_k \triangleq [s_1, s_2, \cdots, s_T]_k$ represents the latent variable in the $k$-th diffusion step, and $\tau_0$ is the original trajectory. A notable property of the forward process is that $\tau_k$ at an arbitrary time-step $k$ can be sampled in closed form as:

$$q(\tau_k|\tau_0) = \mathcal{N}(\tau_k; \sqrt{\bar{\alpha}_k}\tau_0, (1 - \bar{\alpha}_k)I), \tag{16}$$

where $\alpha_k = 1 - \beta_k$ and $\bar{\alpha}_k = \prod_{i=1}^{k} \alpha_k$. When $k \to \infty$, $\tau_k$ approaches a standard Gaussian distribution. In particular, AIGB employs a cosine noise schedule (Nichol & Dhariwal, 2021) to control the schedule $\beta_k$.

**Reverse Process.** In the reverse process, diffusion models aim to remove the added noise on $\tau_K$ and recursively recover $\tau_0$. This process is governed by the conditional model $p_\theta(\tau_{k-1}|\tau_k, y(\tau))$, which is parameterized through a noise prediction model $\epsilon_\theta(\tau_k, y(\tau), k)$. AIGB adopts a temporal U-Net (Ronneberger et al., 2015) for the noise prediction model, a common choice in diffusion-based decision-making methods (Ajay et al., 2023).

## B.1 TRAINING STAGE

The training of the diffusion model is typically formulated as minimizing the mean squared error between the predicted noise $\epsilon_\theta$ and the true noise applied during the forward diffusion process. Specifically, during each iteration, we randomly sample a trajectory from the offline dataset $\mathcal{D}$ and pick a time step $t \in [T]$. We recursively add the Gaussian noise $\epsilon$ to the states in $\tau$ with time steps bigger than $t$ and predict the added noises with $\epsilon_\theta(\tau_k, y(\tau), k)$, where the states between 0 and $t$ in $\tau_k$ are set to real history states $s_1, s_2, \cdots, s_t$. In addition to this standard objective, AIGB also incorporates a supervised loss that measures the discrepancy between the true actions and the actions predicted by an inverse dynamics model $\hat{f}_\phi(s_t, \hat{s}_{t+1})$. Overall, the complete training objective of AIGB can be expressed as:

$$\mathcal{L}(\theta, \phi) = \mathbb{E}_{k,\tau \in \mathcal{D}}[||\epsilon - \epsilon_\theta(\tau_k, y(\tau), k)||^2] + \mathbb{E}_{(s_t, a_t, \hat{s}_{t+1}) \in \mathcal{D}}[||a_t - \hat{f}_\phi(s_t, \hat{s}_{t+1})||^2]. \tag{17}$$

During training, the condition $y(\tau)$ is randomly dropped to enhance model robustness. This technique ensures that both the unconditional model $\epsilon_\theta(\tau_k, k)$ and the conditional model $\epsilon_\theta(\tau_k, y(\tau), k)$ are effectively trained together.

## B.2 INFERENCE STAGE

Starting with Gaussian noise, trajectories are iteratively generated through a series of denoising steps. Specifically, AIGB uses a classifier-free guidance strategy (Ho & Salimans, 2021) to guide the generation of bidding and extract high-likelihood trajectories in the dataset. During generation, AIGB combines conditional and unconditional score estimates linearly:

$$\hat{\epsilon}_k := \epsilon_\theta(\tau_k, k) + \omega\left(\epsilon_\theta\left(\tau_k, y(\tau), k\right) - \epsilon_\theta\left(\tau_k, k\right)\right), \tag{18}$$

where $\omega$ is the guidance scale that controls the influence of the condition $y(\tau)$. This formulation effectively steers the trajectory generation toward regions of the data distribution that are most consistent with the given condition. The predicted state at each step is sampled from $p_\theta(\tau_{k-1}|\tau_k, y(\tau))$:

$$\tau_{k-1} \sim \mathcal{N}\left(\tau_{k-1}|\mu_\theta\left(\tau_k, y(\tau), k\right), \Sigma_\theta\left(\tau_k, k\right)\right), \tag{19}$$

with mean and variance defined as $\mu_\theta(\tau_k, y(\tau), k) = \frac{1}{\sqrt{\alpha_k}}(\tau_k - \frac{\beta_k}{\sqrt{1-\bar{\alpha}_k}}\hat{\epsilon}_k)$ and $\Sigma_\theta(\cdot) = \beta_k$. Note that the initial noisy trajectory $\tau'_K \sim \mathcal{N}(0, I)$ is assigned with history states $s_{1:t}$ for the first $t$ states to ensure history consistency. This is consistent with the training process. By recursively applying the reverse diffusion process using:

$$\tau'_{k-1} = \mu_\theta(\tau'_k, y(\tau), k) + \sqrt{\beta_k}z, \tag{20}$$

where $z \sim \mathcal{N}(0, I)$, we obtain the final denoised trajectory $\tau'_0$, from which the next state $\hat{s}_{t+1}$ is derived. Then the action is generated through an inverse dynamics $\hat{a}_t = \hat{f}_\phi(s_t, \hat{s}_{t+1})$.

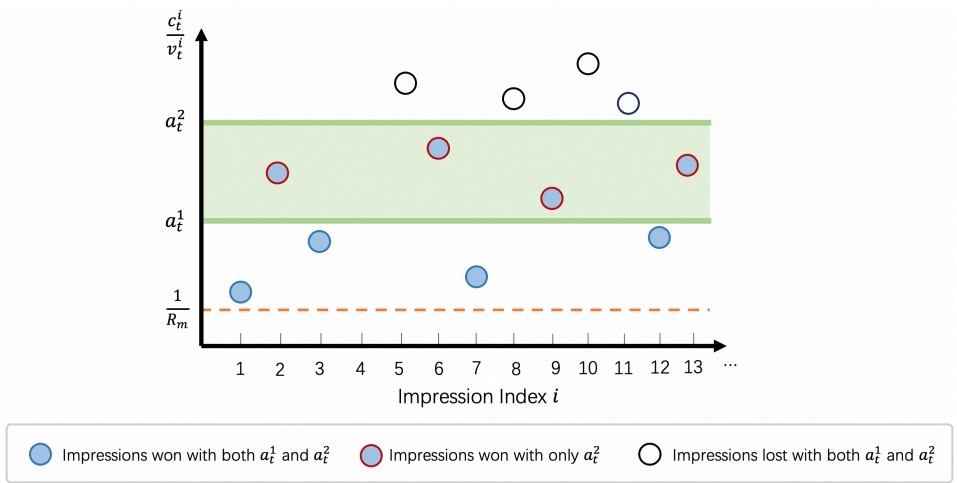

Figure 6: The impression opportunities within time step $t$ and $t+1$, where $p_t^i/v_t^i$ is the $1/\text{ROI}$ of impression $i$. Without loss of generality, consider two actions $a_{1,t}$ and $a_{2,t}$, and let $a_{2,t} \geq a_{1,t}$. The impressions within the shadow area are the impressions won by action $a_{2,t}$ but lost by action $a_{1,t}$.

## C  THEORETICAL PROOFS

### C.1  PROOF OF THEOREM 1.

**Theorem 1** (Lipschitz Continuous of $y(\tau)$.)**.** *The trajectory quality $y(\tau)$ is $\sqrt{T}R_m$-Lipschitz continuous with respect to the Frobenius norm.*

*Proof.* Recall from Section 2.1 that the cost $c_t$ and reward $r_t$ under action $a_t$ between time step $t$ and $t+1$ can be written as:

$$c_t = \sum_i \mathbb{1}\left\{a_t \geq \frac{p_t^i}{v_t^i}\right\}p_t^i \quad \text{and} \quad r_t = \sum_i \mathbb{1}\left\{a_t \geq \frac{p_t^i}{v_t^i}\right\}v_t^i, \tag{21}$$

where $p_t^i$ and $v_t^i$ denote the market price and the value of the $i$-th impression between time step $t$ and $t+1$. Accordingly, the cost ratio $\bar{c}_t$ and the normalized reward $\bar{r}_t$ can be written as:

$$\bar{c}_t = \frac{1}{B}\sum_i \mathbb{1}\left\{a_t \geq \frac{p_t^i}{v_t^i}\right\}p_t^i \quad \text{and} \quad \bar{r}_t = \frac{1}{B}\sum_i \mathbb{1}\left\{a_t \geq \frac{p_t^i}{v_t^i}\right\}v_t^i, \tag{22}$$

Consider two different trajectories $\tau_1$ and $\tau_2$ with actions, cost ratios and normalized rewards sequences $\{a_{1,t}, \bar{c}_{1,t}, \bar{r}_{1,t}\}_{t=1}^T$ and $\{a_{2,t}, \bar{c}_{2,t}, \bar{r}_{2,t}\}_{t=1}^T$, respectively. The trajectory quality gap between $\tau_1$ and $\tau_2$ holds that:

$$|y(\tau_1) - y(\tau_2)| = \left|\sum_t \bar{r}_{1,t} - \sum_t \bar{r}_{2,t}\right| \leq \sum_t |\bar{r}_{1,t} - \bar{r}_{2,t}|. \tag{23}$$

Consider the reward gap between time step $t$ and $t+1$, as shown in Fig.6. Without loss of generality, let $a_{2,t} \geq a_{1,t}$. We have:

$$\begin{aligned}
|\bar{r}_{1,t} - \bar{r}_{2,t}| &= \frac{1}{B}\sum_i \left[\mathbb{1}\left\{a_{2,t} \geq \frac{p_t^i}{v_t^i}\right\} - \mathbb{1}\left\{a_{1,t} \geq \frac{p_t^i}{v_t^i}\right\}\right]v_t^i \\
&= \frac{1}{B}\sum_i \mathbb{1}\left\{a_{2,t} \geq \frac{p_t^i}{v_t^i} \geq a_{1,t}\right\}v_t^i \\
&= \frac{1}{B}\sum_i \mathbb{1}\left\{a_{2,t} \geq \frac{p_t^i}{v_t^i} \geq a_{1,t}\right\}\frac{v_t^i}{p_t^i}p_t^i \\
&\leq \frac{R_m}{B}\sum_i \mathbb{1}\left\{a_{2,t} \geq \frac{p_t^i}{v_t^i} \geq a_{1,t}\right\}p_t^i.
\end{aligned} \tag{24}$$

Note that the cost ratio gap between time step $t$ and $t + 1$ can be written as:

$$|\bar{c}_{1,t} - \bar{c}_{2,t}| = \frac{1}{B} \sum_i \left[ \mathbb{1}\left\{a_{2,t} \geq \frac{p_t^i}{v_t^i}\right\} - \mathbb{1}\left\{a_{1,t} \geq \frac{p_t^i}{v_t^i}\right\} \right] p_t^i = \frac{1}{B} \sum_i \mathbb{1}\left\{a_{2,t} \geq \frac{p_t^i}{v_t^i} \geq a_{1,t}\right\} p_t^i.$$
(25)

Therefore, combining Eq. 24 and Eq. 25, we have:

$$|\bar{r}_{1,t} - \bar{r}_{2,t}| \leq R_m |\bar{c}_{1,t} - \bar{c}_{2,t}|.$$
(26)

We examine the Frobenius norm of the gap between $\tau_1$ and $\tau_2$:

$$
\begin{aligned}
\|\tau_1 - \tau_2\|_F &= \left\| \begin{bmatrix} 1 & \bar{c}_{1,0} & x \\ 2 & \bar{c}_{1,1} & x \\ \vdots & \vdots & \vdots \\ T & c_{1,T-1} & x \end{bmatrix} - \begin{bmatrix} 1 & \bar{c}_{2,0} & x \\ 2 & \bar{c}_{2,1} & x \\ \vdots & \vdots & \vdots \\ T & \bar{c}_{2,T-1} & x \end{bmatrix} \right\|_F \\
&= \sqrt{\sum_t (\bar{c}_{1,t} - \bar{c}_{2,t})^2} \\
&\geq \frac{1}{\sqrt{T}} \sum_t |\bar{c}_{1,t} - \bar{c}_{2,t}| \qquad \text{(Cauchy-Schwarz Inequality)}
\end{aligned}
$$
(27)

Combining Eq. 23, Eq. 26 and Eq. 27, we can obtain that:

$$
\begin{aligned}
|y(\tau_1) - y(\tau_2)| &\leq \sum_t |\bar{r}_{1,t} - \bar{r}_{2,t}| \\
&\leq R_m \sum_t |\bar{c}_{1,t} - \bar{c}_{2,t}| \\
&\leq \sqrt{T} R_m \frac{1}{\sqrt{T}} \sum_t |\bar{c}_{1,t} - \bar{c}_{2,t}| \\
&\leq \sqrt{T} R_m \|\tau_1 - \tau_2\|_F.
\end{aligned}
$$
(28)

This concludes the proof. $\qquad \square$

## C.2 PROOF OF THEOREM 2

Here, we list two lemmas used in the proof of Theorem 2.

**Lemma 1** (Additivity of the Lipschitz). *Let $f_1(x)$ and $f_2(x)$ be two Lipschitz continuous functions with Lipschitz constants $L_1 > 0$ and $L_2 > 0$, respectively. Then $|f_1(x) + f_2(x)|$ is also a Lipschitz continuous function, with Lipschitz constant at most $L_1 + L_2$.*

*Proof.* Recall the Reverse Triangle Inequality states that $\forall a, b$, we have $||a| - |b|| \leq |a - b|$. Then, $\forall x, y$, we have:

$$
\begin{aligned}
||f_1(x) + f_2(x)| - |f_1(y) + f_2(y)|| &\leq |f_1(x) + f_2(x) - f_1(y) - f_2(y)| \\
&\leq |f_1(x) - f_1(y)| + |f_2(x) - f_2(y)| \\
&\leq (L_1 + L_2)|x - y|.
\end{aligned}
$$
(29)

This concludes the proof. $\qquad \square$

**Lemma 2** (Kantorovich-Rubinstein Duality Theorem (Villani, 2021)). *Let $(X, d)$ be a metric space, and let $p$ and $q$ be two probability distributions on $X$. Let $f : X \to \mathbb{R}$ be an $L$-Lipschitz function, and $W_1(p, q)$ denotes the 1-Wasserstein distance between $p$ and $q$. Then we have:*

$$|\mathbb{E}_{x \sim p} f(x) - \mathbb{E}_{x \sim q} f(x)| \leq L \cdot W_1(p, q).$$
(30)

We next give the proof of Theorem 2.

**Theorem 2** (Evaluator Bias in Planning Performance Bound). *Let the upper bound of the evaluator's bias on its training dataset $\mathcal{D}$ be $\delta_D > 0$. The gap between the planner's score $L(\theta)$ and its true performance $J(\theta)$ can be bounded by:*

$$|J(\theta) - L(\theta)| \leq \delta_D + (1+k)\sqrt{T}R_m\mathbb{E}_{y\sim p_D(y)}\bigg[\underbrace{W_1(p_\theta(\tau|y^*), p_\theta(\tau|y))}_{\textit{Lipschitz sensitivity to } y} + \underbrace{W_1(p_\theta(\tau|y), p_D(\tau|y))}_{\textit{imitation error on } \mathcal{D}}\bigg],$$

*where $W_1$ denotes the 1-Wasserstein distance.*

*Proof.* The evaluator bias in the planner's performance can be written as:

$$|J(\theta) - L(\theta)| = |\mathbb{E}_{\tau\sim p_\theta(\tau|y^*)}[y(\tau) - \hat{y}_\phi(\tau)]| \leq \mathbb{E}_{\tau\sim p_\theta(\tau|y^*)}\underbrace{|y(\tau) - \hat{y}_\phi(\tau)|}_{\triangleq f(\tau)} \tag{31}$$

Let $f(\tau) \triangleq |y(\tau) - \hat{y}_\phi(\tau)|$ be the evaluator bias in trajectory $\tau$. From Theorem 1 and Lemma 1, we know that $f(\tau)$ is a $(1+k)\sqrt{T}R_m$-Lipschitz continuous function. Then, we have:

$$
\begin{aligned}
|J(\theta) - L(\theta)| &\leq \mathbb{E}_{\tau\sim p_\theta(\tau|y^*)}f(\tau) \\
&= \mathbb{E}_{y\sim p_D(y)}\bigg[\mathbb{E}_{\tau\sim p_\theta(\tau|y^*)}f(\tau) - \mathbb{E}_{\tau\sim p_D(\tau|y)}f(\tau) + \mathbb{E}_{\tau\sim p_D(\tau|y)}f(\tau)\bigg] \\
&= \underbrace{\mathbb{E}_{y\sim p_D(y)}\mathbb{E}_{\tau\sim p_D(\tau|y)}f(\tau)}_{\leq \delta_D} + \mathbb{E}_{y\sim p_D(y)}\bigg[\mathbb{E}_{\tau\sim p_\theta(\tau|y^*)}f(\tau) - \mathbb{E}_{\tau\sim p_D(\tau|y)}f(\tau)\bigg] \\
&= \delta_D + \mathbb{E}_{y\sim p_D(y)}\underbrace{\bigg[\mathbb{E}_{\tau\sim p_\theta(\tau|y^*)}f(\tau) - \mathbb{E}_{\tau\sim p_\theta(\tau|y)}f(\tau)\bigg]}_{\leq(1+k)\sqrt{T}R_m W_1(p_\theta(\tau|y^*),p_\theta(\tau|y)),\ \text{(Lemma 2)}} \\
&\quad + \mathbb{E}_{y\sim p_D(\tau)}\underbrace{\bigg[\mathbb{E}_{\tau\sim p_\theta(\tau|y)}f(\tau) - \mathbb{E}_{\tau\sim p_D(\tau|y)}f(\tau)\bigg]}_{\leq(1+k)\sqrt{T}R_m W_1(p_\theta(\tau|y),p_D(\tau|y)),\ \text{(Lemma 2)}} \\
&\leq \delta_D + (1+k)\sqrt{T}R_m\mathbb{E}_{y\sim p_D(y)}[W_1(p_\theta(\tau|y^*), p_\theta(\tau|y))] \\
&\quad + (1+k)\sqrt{T}R_m\mathbb{E}_{y\sim p_D(y)}[W_1(p_\theta(\tau|y), p_D(\tau|y))]. \tag{32}
\end{aligned}
$$

Therefore, we have:

$$|J(\theta) - L(\theta)| \leq \delta_D + (1+k)\sqrt{T}R_m\mathbb{E}_{y\sim p_D(y)}\bigg[W_1(p_\theta(\tau|y^*), p_\theta(\tau|y)) + W_1(p_\theta(\tau|y), p_D(\tau|y))\bigg]. \tag{33}$$

This concludes the proof. $\qquad\square$

### C.3 PROOF OF EQ. 6

We give the proof of Eq. 6 as follows. Denote $\text{Lip}_{W_1}(p_\theta(\tau|y))$ as the planner's Lipschitz constant with respect to $y$ regarding the Wasserstein distance $W_1$, we have:

$$
\begin{aligned}
\mathbb{E}_{y\sim p_D(y)}[W_1(p_\theta(\tau|y^*), p_\theta(\tau|y))] &\leq \text{Lip}_{W_1}(p_\theta(\tau|y))\mathbb{E}_{y\sim p_D(y)}[((1+\epsilon)y_m - y)] \\
&= \text{Lip}_{W_1}(p_\theta(\tau|y))\int_0^{y_m} p_D(y)[(1+\epsilon)y_m - y]\mathrm{d}y \\
&\leq \text{Lip}_{W_1}(p_\theta(\tau|y))\int_0^{y_m} p_D(y)[(1+\epsilon)y_m]\mathrm{d}y \\
&= (1+\epsilon)y_m\text{Lip}_{W_1}(p_\theta(\tau|y)), \tag{34}
\end{aligned}
$$

where we leverage the non-negativity property of the condition $y \geq 0, \forall y \in \mathcal{D}$. This completes the proof.

## C.4  PROOF OF EQ. 7

**Lemma 3** (Pinsker's Inequality (Tsybakov, 2008)). *Let $P$ and $Q$ be two probability measures defined on the same measurable space, and assume that $P$ is absolutely continuous with respect to $Q$, i.e., $P \ll Q$. Then the total variation distance between $P$ and $Q$ is bounded above by the KL divergence from $P$ to $Q$ as follows:*

$$\|P - Q\|_{TV} \le \sqrt{\frac{1}{2}D_{KL}(P\|Q)}. \tag{35}$$

**Lemma 4** (Wasserstein–Total Variation Inequality on Bounded Metric Spaces (Villani et al., 2008)). *Let $(\mathcal{Z}, d)$ be a metric space with diameter $diam(\mathcal{Z}) \triangleq \sup_{z_1, z_2 \in \mathcal{Z}} d(z_1, z_2)$. Let $P$ and $Q$ be two probability measures on $\mathcal{Z}$. Then the 1-Wasserstein distance between $P$ and $Q$ satisfies:*

$$W_1(P, Q) \le diam(\mathcal{Z})\|P - Q\|_{TV}. \tag{36}$$

We give the proof of Eq. 7 as follows. Equipped with the above two lemmas, we have:

$$W_1(p_\theta(\tau|y), p_D(\tau|y)) \le \text{diam}(\mathcal{T})\|p_\theta(\tau|y) - p_D(\tau|y)\|_{\text{TV}}$$

$$\le \text{diam}(\mathcal{T})\sqrt{\frac{1}{2}D_{KL}(p_D(\tau|y)\|p_\theta(\tau|y))}, \tag{37}$$

where $\mathcal{T}$ is the trajectory space. Note that due to the budget constraint $\sum_t c_t \le B$ [6], we have the sum of the cost ratio satisfies $\sum_t \bar{c}_t \le 1$. The trajectory space can be expressed as:

$$\mathcal{T} = \left\{ \Big[[1, \bar{c}_0, x], [2, \bar{c}_1, x], \cdots, [T, \bar{c}_{T-1}, x]\Big] \Big| \bar{c}_t \ge 0, \forall t, \text{ and } \sum_t \bar{c}_t \le 1 \right\} \tag{38}$$

We next prove that the diameter of the trajectory space, $\text{diam}(\mathcal{T})$, can be bounded by a constant. Specifically, the diameter only depends on the largest possible distance between the cost ratio sequences in two trajectories since:

$$\begin{aligned}
\text{diam}(\mathcal{T}) &= \sup_{\tau_1, \tau_2 \in \mathcal{T}} \|\tau_1 - \tau_2\|_F \\
&= \sup_{\tau_1, \tau_2 \in \mathcal{T}} \left\| \begin{bmatrix} 1 & \bar{c}_{1,0} & x \\ 2 & \bar{c}_{1,1} & x \\ \vdots & \vdots & \vdots \\ T & c_{1,T-1} & x \end{bmatrix} - \begin{bmatrix} 1 & \bar{c}_{2,0} & x \\ 2 & \bar{c}_{2,1} & x \\ \vdots & \vdots & \vdots \\ T & \bar{c}_{2,T-1} & x \end{bmatrix} \right\|_F \\
&= \sup_{\tau_1, \tau_2 \in \mathcal{T}} \sqrt{\sum_t (\bar{c}_{1,t} - \bar{c}_{2,t})^2}. \tag{39}
\end{aligned}$$

For convenience, we let $\mathbf{c}_i \triangleq [\bar{c}_{i,0}, \bar{c}_{i,2}, \cdots, \bar{c}_{i,T-1}], i \in \{1, 2\}$. Then, the key part in the above result, $\sum_t (\bar{c}_{1,t} - \bar{c}_{2,t})^2$, can be written as:

$$\begin{aligned}
\sum_t (\bar{c}_{1,t} - \bar{c}_{2,t})^2 &= \sum_t (\bar{c}_{1,t}^2 - 2\bar{c}_{1,t}\bar{c}_{2,t} + \bar{c}_{2,t}^2) \\
&= \|\mathbf{c}_1\|_2^2 + \|\mathbf{c}_2\|_2^2 - 2\langle \mathbf{c}_1, \mathbf{c}_2 \rangle \\
&\le \|\mathbf{c}_1\|_2^2 + \|\mathbf{c}_2\|_2^2, \tag{40}
\end{aligned}$$

where $\langle \mathbf{c}_1, \mathbf{c}_2 \rangle \ge 0$. As $\bar{c}_{i,t} \ge 0$ and $\sum_t \bar{c}_{i,t} \le 1$, we have $0 \le \bar{c}_{i,t} \le 1$. Therefore, it holds that:

$$\|\mathbf{c}_i\|_2^2 = \sum_t \bar{c}_{i,t}^2 \le \sum_t \bar{c}_{i,t} \le 1. \tag{41}$$

Combining Eq. 40 and Eq. 41, we have:

$$\sqrt{\sum_t (\bar{c}_{1,t} - \bar{c}_{2,t})^2} \le \sqrt{\|\mathbf{c}_1\|_2^2 + \|\mathbf{c}_2\|_2^2} \le \sqrt{2}. \tag{42}$$

---

[6]As explained in Footnote 1, the budget constraint is guaranteed to be satisfied in real-world advertising systems thanks to an automatic suspension mechanism that halts bidding once the budget is exhausted.

Therefore, we have $\text{diam}(\mathcal{T}) = \sqrt{2}$. According to Eq. 37, we have:

$$W_1(p_\theta(\tau|y), p_D(\tau|y)) \leq \sqrt{D_{KL}(p_D(\tau|y)\|p_\theta(\tau|y))}. \tag{43}$$

Recall that we impose the KL-constraint as:

$$\mathbb{E}_{y \sim p_D(y)}[D_{\text{KL}}(p_D(\tau|y)\|p_\theta(\tau|y))] \leq \delta_K, \tag{44}$$

Taking the expectation over $y \sim p_D(y)$ on both sides of Eq. 43, we have:

$$\mathbb{E}_{y \sim p_D(y)}[W_1(p_\theta(\tau|y), p_D(\tau|y))] \leq \mathbb{E}_{y \sim p_D(y)}\left[\sqrt{D_{KL}(p_D(\tau|y)\|p_\theta(\tau|y))}\right]$$

$$\leq \sqrt{\mathbb{E}_{y \sim p_D(y)}[D_{KL}(p_D(\tau|y)\|p_\theta(\tau|y))]} \quad \text{(Jensen Inequality)}$$

$$\leq \sqrt{\delta_K} \quad \text{(KL constraint Eq.44).} \tag{45}$$

This completes the proof.

## C.5   PROOF OF THEOREM 3

**Theorem 3** (Sub-optimality Gap Bound). *Let $\delta_M \triangleq \mathbb{E}_{y \sim p_D(y)}[D_{KL}(p_D(\tau|y)\|p_{\theta^*}(\tau|y))]$ be the expected distance between the optimal trajectory distribution and the trajectory distribution of the offline dataset $\mathcal{D}$. The true performance gap between the optimal parameter $\theta^*$ and the solution $\hat{\theta}$ to Eq. 8 is bounded by:*

$$J(\theta^*) - J(\hat{\theta}) \leq 2\delta_D + (1+2k)\sqrt{T}R_m\left[\sqrt{\delta_M} + \sqrt{\delta_K} + (1+\epsilon)y_m L_p\right]. \tag{46}$$

*Proof.* The sub-optimality gap can be expressed as follows:

$$J(\theta^*) - J(\hat{\theta}) = \left(J(\theta^*) - L(\theta^*)\right) + \left(L(\theta^*) - L(\hat{\theta})\right) + \left(L(\hat{\theta}) - J(\hat{\theta})\right)$$

$$\leq \underbrace{|J(\theta^*) - L(\theta^*)|}_{\text{evaluator bias in } p_{\theta^*}} + \underbrace{|L(\theta^*) - L(\hat{\theta})|}_{\text{score gap}} + \underbrace{|L(\hat{\theta}) - J(\hat{\theta})|}_{\text{evaluator bias in } p_{\hat{\theta}}}. \tag{47}$$

We examine the above three terms accordingly.

**(1) Evaluator Bias in $p_{\theta^*}$.** Denote the evaluator bias on trajectory $\tau$ as $f(\tau) \triangleq |y(\tau) - \hat{y}_\phi(\tau)|$. Following the derivation process in Eq. 32, we have:

$$|J(\theta^*) - L(\theta^*)| \leq \mathbb{E}_{y \sim p_D(y)}\mathbb{E}_{\tau \sim p_D(\tau|y)}f(\tau) + \mathbb{E}_{y \sim p_D(y)}\left[\mathbb{E}_{\tau \sim p_{\theta^*}(\tau|y^*)}f(\tau) - \mathbb{E}_{\tau \sim p_D(\tau|y)}f(\tau)\right]$$

$$\leq \delta_D + (1+k)\sqrt{T}R_m\mathbb{E}_{y \sim p_D(y)}[W_1(p_{\theta^*}(\tau|y^*), p_D(\tau|y))], \tag{48}$$

where $W_1(p_{\theta^*}(\tau|y^*), p_D(\tau|y))$ denotes the probability distribution distance between the optimal planner and the offline dataset. Based on the derivation in Appendix C.4, we have:

$$\mathbb{E}_{y \sim p_D(y)}[W_1(p_{\theta^*}(\tau|y^*), p_D(\tau|y))] \leq \sqrt{\mathbb{E}_{y \sim p_D(y)}[D_{KL}(p_D(\tau|y)\|p_{\theta^*}(\tau|y^*))]} \tag{49}$$

Let $\delta_M \triangleq \mathbb{E}_{y \sim p_D(y)}[D_{KL}(p_D(\tau|y)\|p_{\theta^*}(\tau|y^*))]$ be the distance between the optimal trajectory distribution and the offline dataset trajectory distribution. We have:

$$|J(\theta^*) - L(\theta^*)| \leq \delta_D + (1+k)\sqrt{T}R_m\sqrt{\delta_M}. \tag{50}$$

**(2) Score Gap.** Recall that the trained evaluator $\hat{y}_\phi(\tau)$ is a $k\sqrt{T}R_m$-Lipschitz continuous function with the Lipschitz constraint design. With Lemma 2, we have:

$$|L(\theta^*) - L(\hat{\theta})| = |\mathbb{E}_{\tau \sim p_{\theta^*}(\tau|y^*)}\hat{y}_\phi(\tau) - \mathbb{E}_{\tau \sim p_{\hat{\theta}}(\tau|y^*)}\hat{y}_\phi(\tau)|$$

$$\leq k\sqrt{T}R_m W_1(p_{\theta^*}(\tau|y^*), p_{\hat{\theta}}(\tau|y^*))$$

$$\leq k\sqrt{T}R_m\mathbb{E}_{y \sim p_D(y)}\left[W_1(p_{\theta^*}(\tau|y^*), p_D(\tau|y)) + W_1(p_D(\tau|y), p_{\hat{\theta}}(\tau|y^*))\right]$$

$$\leq k\sqrt{T}R_m\left[\sqrt{\delta_M} + (1+\epsilon)y_m L_p + \sqrt{\delta_K}\right], \tag{51}$$

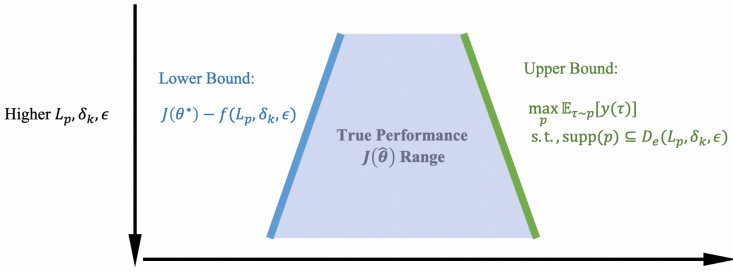

Figure 7: Performance range of the planner with respect to hyper-parameters $\delta_k, L_p$ and $\epsilon$. There is a trade-off in the selection of hyper-parameters: larger $\delta_k, L_p, \epsilon$ results in a smaller lower bound but a higher upper bound.

where we leverage the fact that $\hat{\theta}$ is a solution to Eq. 8 which satisfies the KL and Lipschitz constraint, and we leverage the results in Eq. 6, Eq. 7 and Eq. 49.

**(3) Evaluator Bias in $p_{\hat{\theta}}$.** Since $\hat{\theta}$ satisfies the KL and Lipschitz constraint in Eq. 8, we can use the results in Theorem 2, Eq. 6 and Eq. 7 to obtain:

$$|J(\hat{\theta}) - L(\hat{\theta})| \leq \delta_D + (1+k)\sqrt{T}R_m[(1+\epsilon)y_m L_p + \sqrt{\delta_K}], \tag{52}$$

Overall, combining the results in Eq. 50, Eq. 51, and Eq. 52, we have:

$$J(\theta^*) - J(\hat{\theta}) \leq 2\delta_D + (1+2k)\sqrt{T}R_m\left[\sqrt{\delta_M} + \sqrt{\delta_K} + (1+\epsilon)y_m L_p\right]. \tag{53}$$

This concludes the proof.

$\square$

### C.6 PROOF OF SCORE GRADIENT

The probability of the trajectory generated by the Causal Transformer can be decomposed into:

$$p_\theta(s_{1:T}|y) = \Pi_t p_\theta(s_t|s_{1:t-1}, y). \tag{54}$$

Then, we have:

$$
\begin{aligned}
\nabla_\theta L(\theta) &= \nabla_\theta \int_\tau p_\theta(\tau|y^*)\hat{y}_\phi(\tau)\mathrm{d}\tau \\
&= \nabla_\theta \int_{s_1,\cdots,s_T} p_\theta(s_{1:T}, \tau|y^*)\hat{y}_\phi(\tau)\mathrm{d}s_1\cdots\mathrm{d}s_T \\
&= \int_{s_{1:T}} p_\theta(s_{1:T}|y^*)\frac{\nabla_\theta p_\theta(s_{1:T}|y^*)}{p_\theta(s_{1:T}|y^*)}\hat{y}_\phi(\tau)\mathrm{d}s_1\cdots\mathrm{d}s_T \\
&= \mathbb{E}_{s_{1:T}\sim p_\theta(s_{1:T}|y^*)}\left[\nabla_\theta \log p_\theta(s_{1:T}|y^*)\hat{y}_\phi(\tau)\right] \\
&= \mathbb{E}_{s_{1:T}\sim p_\theta(s_{1:T}|y^*)}\left[\nabla_\theta \log \Pi_t p_\theta(s_t|s_{1:t-1}, y^*)\hat{y}_\phi(\tau)\right] \\
&= \mathbb{E}_{s_{1:T}\sim p_\theta(s_{1:T}|y^*)}\left[\sum_t \nabla_\theta \log p_\theta(s_t|s_{1:t-1}, y^*)\hat{y}_\phi(\tau)\right]. 
\end{aligned}
\tag{55}
$$

## D THEORETICAL PERFORMANCE RANGE TRADEOFF DISCUSSION

We note that Theorem 3 actually gives the lower bound of the planner's true performance $J(\hat{\theta})$, i.e.,

$$J(\theta^*) - \underbrace{\left(2\delta_D + (1+2k)\sqrt{T}R_m\left[\sqrt{\delta_M} + \sqrt{\delta_K} + (1+\epsilon)y_m L_p\right]\right)}_{\triangleq f(\delta_K, L_p, \epsilon)} \leq J(\hat{\theta}), \tag{56}$$

where we denote that gap term $f$ as a function of hyper-parameters $\delta_K, L_p$ and $\epsilon$ that we can adjust. As $f(\delta_K, L_p, \epsilon)$ is monotonically increasing with respect to $\delta_K, L_p$ and $\epsilon$, smaller values of these terms result in a higher lower bound of $J(\hat{\theta})$. However, as illustrated in Fig. 1, these three terms also determine the planner's exploration range, denoted as $D_e(\delta_k, L_p, \epsilon)$. Specifically, higher values of $\delta_k, L_p, \epsilon$ indicate a larger exploration range, which thereby leads to a higher performance upper bound of the planner, i.e.,

$$J(\hat{\theta}) \leq \max_{p(\tau)} \mathbb{E}_{\tau \sim p(\tau)}[y(\tau)] \quad \text{s.t.,} \ \ \text{supp}(p(\tau)) \subseteq D_e(\delta_k, L_p, \epsilon). \tag{57}$$

This introduces a trade-off in the selection of hyper-parameters. As illustrated in Fig. 7, higher $\delta_k, L_p, \epsilon$ results in a smaller lower bound but a higher upper bound. To this end, we conduct hyper-parameter tuning and give hyper-parameter determination in Appendix F.8.

# E   AIGB-PEARL ALGORITHM SUMMARY

Algorithm 1 summarizes the training process of AIGB-Pearl. Specifically, we compute the Lipschitz constants of $y(\tau)$ and $p_D(\tau|y)$ using the offline dataset $\mathcal{D}$ and, accordingly, determine the Lipschitz constraints for the evaluator and the planner, respectively. Then, we sequentially perform evaluator learning, planner pretraining, and KL-Lipschitz-constrained score maximization for the planner. The development of AIGB-Pearl is supported by ROLL (Wang et al., 2025).

## E.1   ADDITIONAL DESIGNS FOR EVALUATOR ACCURACY ENHANCEMENT

To further enhance the reliability of the trajectory evaluator, we design two additional techniques for the evaluator learning. Specifically, as described in the following, we (i) integrate an LLM embedding into its input feature for better representational capacity; and (ii) employ the pair-wise loss for better score estimation accuracy. Fig. 8 illustrates the complete evaluator learning losses. The effectiveness of these two methods is studied in Appendix F.6.

### E.1.1   LLM EMBEDDING ENHANCEMENT

Motivated by the success of integrating user-specific features into recommendation systems (Chen et al., 2015), we incorporate advertiser-specific features into the trajectory evaluator to enhance its representational capacity and improve scoring accuracy. Note that some advertiser-specific features are textual in nature—such as product titles, categories, and reviews—and are therefore difficult to incorporate directly into the vectorized trajectory $\tau$. To address this, we construct a *prompt* containing all such textual attributes and employ a pre-trained large language model (LLM), QWen2.5-1.5B-Instruct, with general world knowledge to extract $T$ representation, which we refer to as the *LLM embedding*. Specifically, we use the output of the last hidden layer as the LLM embedding. This embedding is then used as an additional positional encoding in the evaluator. The prompt template is given below:

---

**LLM Prompt Template.** I am an [*advertising platform*] advertiser, operating the [*brand name*] brand in the category of [*category name*], classified as a [*advertiser tier*] tier advertiser. I have a product titled [*product name*] currently running in advertising campaigns. This product belongs to the leaf category of [*leaf category*], priced at [*product price*], and is positioned in the [*price range*]. Its price ranks within the top [*price ranking in the leaf category*] % in the leaf category.

Historical Average Performance: The product generates an average of [*average daily transactions*] daily transactions, with a GMV of [*average daily GMV*], driven by advertising. It receives an average [*average daily impressions*] daily impressions from search and recommendation traffic, [*click numbers*] clicks, [*average daily BuyCnt*] BuyCnt, and a GMV of [*GMV*]. It ranks within the top [*sales volume ranking in leaf category*] % in sales volume in the leaf category, with an average transaction value of [*average transaction value*].

Historical Time-based Average Performance: This product has undergone continuous exposure to advertising for *number of advertising days* days. The average hourly advertising spend distribution per day (from 0:00 to 24:00) is *historical spend distribution sequence*. The average GMV distribution across its category during this period is *historical GMV distribution sequence*. The average daily advertising budget is *daily advertising budget*.

---

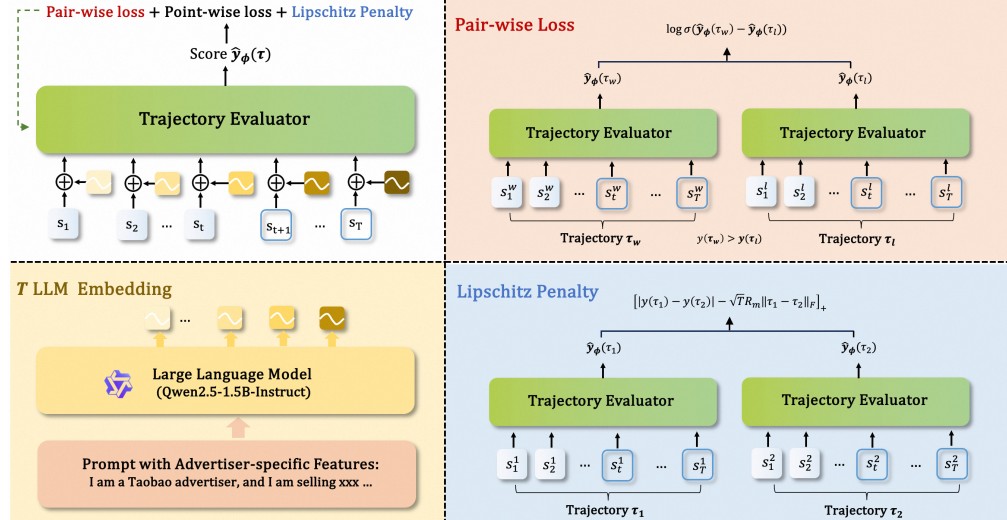

Figure 8: **AIGB-Pearl Evaluator Learning.** The evaluator loss is composed of three parts, including the point-wise loss, the pair-wise loss, and the Lipschitz penalty. The evaluator's input representation is augmented with an LLM embedding to incorporate semantic information from the advertiser's textual features.

### E.1.2 PAIR-WISE LOSS

Unlike human feedback scores used in LLM post-training (Christiano et al., 2017)—which can suffer from subjective biases in their absolute valuations—the trajectory quality $y(\tau)$ has real physical meaning and is comparable among different trajectories. Therefore, we can adopt a hybrid point-wise and pair-wise loss for the score to capture the absolute value of $y(\tau)$ and their relative preference, respectively. This approach has demonstrated superior performance in recommender systems (Cao et al., 2007; Lei et al., 2017; Wang et al., 2022a). Specifically, the pair-wise loss function can be expressed as:

$$\mathbb{E}_{(\tau_w, \tau_l) \sim \mathcal{D}_p}\left[\log \sigma(\hat{y}_\phi(\tau_w) - \hat{y}_\phi(\tau_l))\right], \tag{58}$$

where the pair-wise loss is implemented by the typical Bradley-Terry (BT) function (Bradley & Terry, 1952). Here, $\mathcal{D}_p = \{(\tau_w, \tau_l, y(\tau_w), y(\tau_l))\}$ denotes a pair-wise dataset extracted from the offline dataset $\mathcal{D}$, where $\tau_w$ denotes the trajectory with higher trajectory quality, i.e., $y(\tau_w) \geq y(\tau_l)$.

### E.1.3 OVERALL EVALUATOR LOSS

Combining the previous techniques, the overall evaluator loss is:

$$l_e(\phi) = \underbrace{\mathbb{E}_{\tau \sim \mathcal{D}}\left[(\hat{y}_\phi(\tau) - y(\tau))^2\right]}_{\text{point-wise loss}} + \beta_4 \underbrace{\mathbb{E}_{(\tau_w, \tau_l) \sim \mathcal{D}_p}\left[\log \sigma(\hat{y}_\phi(\tau_w) - \hat{y}_\phi(\tau_l))\right]}_{\text{pair-wise loss}}$$

$$+ \beta_1 \underbrace{\mathbb{E}_{\tau_1, \tau_2}\left[|\hat{y}_\phi(\tau_1) - \hat{y}_\phi(\tau_2)| - \sqrt{T} R_m \|\tau_1 - \tau_2\|_F\right]_+}_{\text{Lipschitz penalty}}, \tag{59}$$

where $\beta_4 > 0$ is a hyper-parameter.

## F ADDITIONAL EXPERIMENTS

### F.1 SIMULATED EXPERIMENT SETTINGS

We include the detailed simulated experiment settings in Table 6. Specifically, we consider the bidding process in a day, where the bidding episode is divided into 96 time steps. Thus, the duration

---

**Algorithm 1:** AIGB-Pearl (Planning with EvaluAtor via RL)

---

**Input**        : Offline dataset $\mathcal{D}$, desired condition $y^*$, hyper-parameters $\beta_1, \beta_2, \beta_3$.
**Output**       : Optimized $\theta$ and $\phi$
**Initialization:** randomly initialized planner parameter $\theta$, trajectory evaluator parameters $\phi$
// Determining the Lipschitz Value
Calculate the Lipschitz value of $y(\tau)$ and $p_D(\tau|y)$ using the offline dataset $\mathcal{D}$.
Set the Lipschitz constraint value $L_e$ for the evaluator and $L_p$ for the planner to be bigger than
   the Lipschitz value of $y(\tau)$ and $p_D(\tau|y)$, respectively.
// Training the trajectory evaluator
**while** *not converged* **do**
 | Update $\phi$ by minimizing Eq. 11;
**end**
// Training the generative planner
Warm start with pretrained planner $p_\theta$;
**while** *not converged* **do**
   | Generate bidding trajectories $\tau \sim p_\theta(\tau|y^*)$;
   | Score generated trajectories with frozen $\phi$: $\hat{y}_\phi(\tau)$;
   | Update $\theta$ by maximizing Eq. 12.
**end**

---

Table 6: Settings of the simulated experiments.

| Parameters | Values |
|---|---|
| Number of advertisers | 30 |
| Time steps in an episode, $T$ | 96 |
| Minimum number of impressions within a time step | 50 |
| Maximum number of impressions within a time step | 300 |
| Minimum budget | 1000 Yuan |
| Maximum budget | 4000 Yuan |
| Value of impressions | $> 0$ |
| Minimum bid price, $\min\{av_i\}$ | 0 Yuan |
| Maximum bid price, $\max\{av_i\}$ | 10 Yuan |
| Maximum market price, $p_M$ | 10 Yuan |

between two adjacent time steps $t$ and $t + 1$ is 15 minutes. The number of impression opportunities between time steps $t$ and $t + 1$ fluctuates from 100 to 500. The minimum and maximum budgets of advertisers are 1000 Yuan and 4000 Yuan, respectively. The upper bound of the bid price is 10 Yuan, and the values of impressions are positive.

**Hardware Resource.** The simulated experiments are conducted based on an NVIDIA T4 Tensor Core GPU. We use 10 CPUs and 200G memory.

## F.2    REAL-WORLD EXPERIMENT SETTINGS

We include the detailed real-world experiment settings in Table 7. Specifically, we consider the bidding process in a day, where the bidding episode is divided into 96 time steps. Thus, the duration between two adjacent time steps $t$ and $t + 1$ is 15 minutes. The number of impression opportunities between time steps $t$ and $t + 1$ fluctuates from 100 to 2,500. The minimum and maximum budgets of advertisers are 50 Yuan and 10,000 Yuan, respectively. The upper bound of the bid price is 25 Yuan, and the values of impressions are positive.

**Hardware Resource.** The training process in the real-world experiments is conducted using 10 NVIDIA T4 Tensor Core GPUs in a distributed manner. For each distributional worker, we use 10 CPUs and 200 GB of memory.

Table 7: Settings of the real-world experiments.

| Parameters | Values |
|---|---|
| Number of advertisers | 6,000 |
| Time steps in an episode, $T$ | 96 |
| Minimum number of impressions within a time step | 100 |
| Maximum number of impressions within a time step | 2,500 |
| Minimum budget | 50 Yuan |
| Maximum budget | 10,000 Yuan |
| Value of impressions | $> 0$ |
| Minimum bid price, $\min\{av_i\}$ | 0 Yuan |
| Maximum bid price, $\max\{av_i\}$ | 25 Yuan |
| Maximum market price, $p_M$ | 25 Yuan |

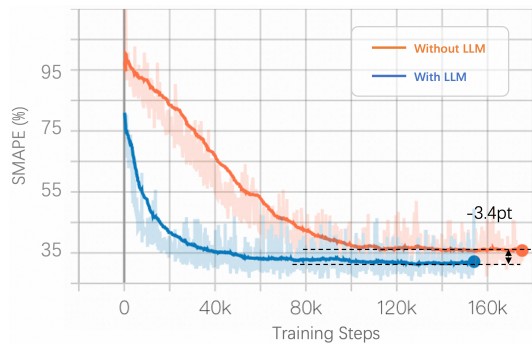

Figure 9: The SMAPE training curves of the trajectory evaluator with and without LLM Embeddings. Incorporating LLM embeddings helps to achieve faster convergence and improved absolute accuracy.

### F.3 REAL-WORLD EXPERIMENTS ON TARGETROAS BIDDING PROBLEM

In addition to the budget-constrained auto-bidding problem, we also apply the proposed AIGB-Pearl algorithm to a more challenging type of auto-bidding problem, named TargetROAS, with an extra ROI (Return on Investment) constraint. We evaluate our method in a real-world experiment on TaoBao involving 300k advertisers over 22 days. The offline dataset comprises 16 million trajectories of 800k advertisers. The results are given in Table 8. AIGB-Pearl achieves a **+5.1%** improvement in GMV compared to the SOTA auto-bidding method, DiffBid, demonstrating its effectiveness in managing more complex and realistic constraints.

### F.4 PATHOLOGICAL TRAJECTORY BEHAVIOR EXPLANATION

In industrial practice, stable and effective metrics have been developed to evaluate pathological behaviors. For the case of the budget-constrained auto-bidding problem with bidding cycles structured as 24-hour episodes (T = 96 time steps), the following three key metrics are commonly used to identify pathological behaviors:

- **Excessive budget consumption**: there exists a time step $t$ such that the cost between time step $t$ and $t + 1$ exceeds $10\%$ of the budget $B$;

- **Forward- (or Backward-) trending pacing**: the cost between time step 1 and 24 (or between time step $T - 24$ and $T$) exceeds $40\%$ (or $40\%$) of the budget $B$;

- **under-utilization of available budgets**: the total cost over the bidding episode is lower than $90\%$ of the budget $B$.

Table 8: Overall performance of TargetROAS in real-world A/B test, involving 300k advertisers over 22 days.

| Methods | GMV | BuyCnt | ROI | Cost |
|---|---|---|---|---|
| **DiffBid** | 779,642,891 | 11,519,082 | 4.68 | 166,544,918 |
| **AIGB-Pearl (ours)** | 819,550,812 | 11,886,501 | 4.70 | 174,234,673 |
| **Δ** | **+5.1%** | **+3.2%** | **+0.5%** | **+4.6%** |

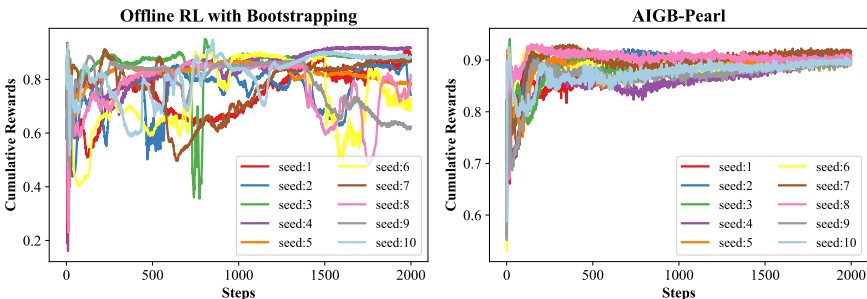

Figure 10: Learning curves of cumulative rewards between offline RL with bootstrapping method and AIGB-Peral under 10 seeds.

## F.5 TRAINING STABILITY

We present additional comparisons between the training curves of the offline RL with bootstrapping and those of AIGB-Pearl in Fig. 10, Fig. 11, Fig. 12, and Fig. 13 concerning:

- **Cumulative Rewards**: the main performance index of the considered auto-bidding problem;
- **Online Rate**: the ratio between the bidding period before the budget runs out and the total bidding period. A larger Online Rate indicates a better performance.
- **Bad Case Rate**: the ratio between the number of "bad" trajectories and the total number of generated trajectories. A lower Bad Case Rate indicates a better performance.
- **Cost Rate**: the ratio between the cost and the budget. A larger cost rate indicates a better performance.

It can be observed that the offline RL method tends to exhibit significant instability throughout training, with high variance across different seeds. In contrast, AIGB-Pearl achieves much smoother and more consistent learning progress, demonstrating the improved training stability.

## F.6 EVALUATOR ACCURACY

**Accuracy Metric.** We evaluate the accuracy of the evaluator along two dimensions, including the *absolute accuracy*, reflecting how close the predicted scores are to ground truth qualities, and the *order accuracy*, reflecting the correctness of relative rankings between trajectory pairs. Specifically, we use the *symmetric mean absolute percentage error*, **SMAPE**, as the metric for the absolute accuracy and the **AUC**, defined as the ratio of correctly predicted ordinal pairs to the total number of pairs, as the metric for the order accuracy. The SMAPE ranges from $0\%$ to $200\%$, and the AUC ranges from $0\%$ to $100\%$. Lower SMAPE and larger AUC indicate better evaluator accuracy.

In the following, we investigate the effectiveness of using LLM embeddings and a pairwise loss for evaluator learning.

**LLM Embedding Effectiveness.** We examine the accuracy of the trajectory evaluator without using LLM embeddings, and the results are reported in the lines of "w/o LLM" in Table 9 and Table 10. It can be observed that LLM embeddings can improve both absolute and order accuracy. Fig. 9 compares the training progress with and without LLM embeddings in terms of the SMAPE

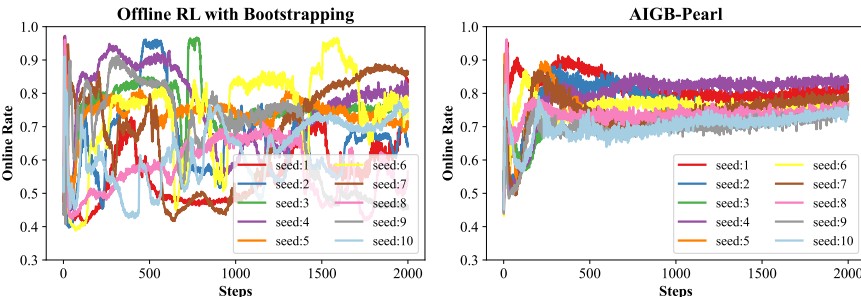

Figure 11: Learning curves of online rate between offline RL with bootstrapping method and AIGB-Peral under 10 seeds.

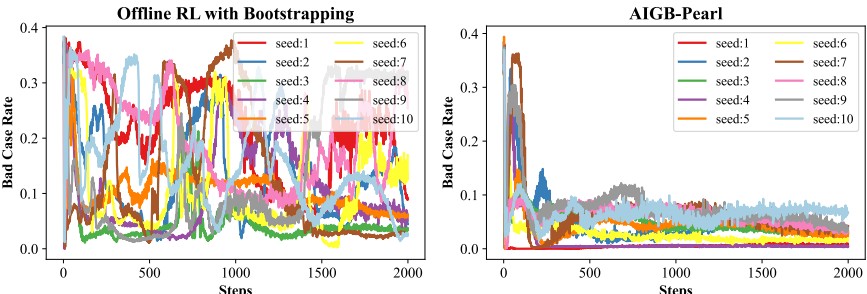

Figure 12: Learning curves of bad case rate between offline RL with bootstrapping method and AIGB-Peral under 10 seeds.

metric. As illustrated, the evaluator incorporating LLM embeddings converges faster and achieves a lower SMAPE than the one without LLM embeddings. The performance gain stems from LLM embeddings' ability to encode high-level semantic information, thereby facilitating a more nuanced understanding of dependencies among sequential bidding states.

**Hybrid Point-wise and Pair-wise Losses Effectiveness.** Table 9 and Table 10 present the SMAPE and AUC of the trajectory evaluator when trained with point-wise loss only, pair-wise loss only, and a combination of both. It can also be seen that using only pairwise loss yields significantly worse SMAPE performance, despite some improvement in AUC. This suggests that while pairwise loss can enhance ranking consistency, it falls short of providing accurate absolute-value predictions. When both point-wise and pair-wise losses are used together, the evaluator achieves lower SMAPE and higher AUC. This indicates that combining these two loss types not only improves absolute accuracy but also enhances order accuracy in trajectory evaluation.

## F.7 EMPIRICAL PERFORMANCE WITH GENERAL OFFLINE DATA DISTRIBUTIONS

Note that in many real-world auto-bidding systems, including the one considered in the paper, due to operational safety constraints, the online-deployed bidding policy is typically a single fixed model, and the offline dataset is collected exclusively from this single policy over multiple days, where an advertiser contributes a single trajectory per day. For example, in the considered auto-bidding system, the online-deployed baseline policy is a conditional generative model that, given a given advertiser and identical conditions, generates identical trajectory plans each day. The variation across

Table 9: SMAPE results from ablation experiments on the trajectory evaluator.

| SMAPE ↓ | Point-wise | Pair-wise | Both |
|---|---|---|---|
| w/o LLM | 43.55% | 196.38% | 35.00% |
| with LLM | 38.20% | 196.00% | **31.60**% |

Table 10: AUC results from ablation experiments on the trajectory evaluator.

| AUC ↑ | Point-wise | Pair-wise | Both |
|---|---|---|---|
| w/o LLM | 61.57% | 64.91% | 70.91% |
| with LLM | 65.00% | 71.30% | **75.10**% |

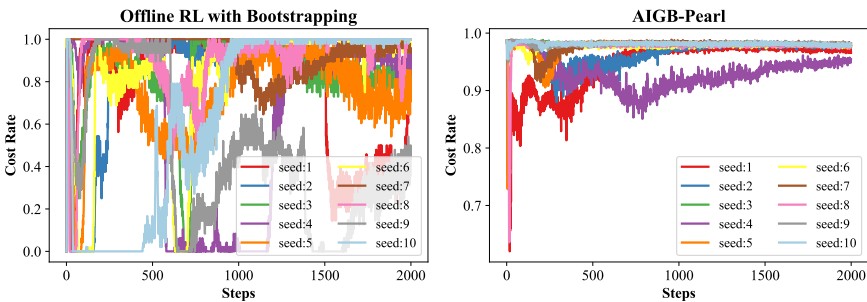

Figure 13: Learning curves of cost rate between offline RL with bootstrapping method and AIGB-Pearl under 10 seeds.

different trajectories of the same advertiser in the offline dataset is solely due to stochastic environmental factors (e.g., traffic fluctuations). Since these exogenous perturbations are typically the sum of many independent impression-level sources of noise, the resulting trajectory deviations can be reasonably approximated as a Gaussian distribution.

To demonstrate the broad applicability of the proposed algorithm, we evaluate its performance in settings where multiple policies are used for data collection. Specifically, we collect trajectories in the simulated environment using nine distinct bidding policies, thereby constructing an offline dataset with a multi-modal distribution that violates the Gaussian distribution. The empirical results are presented in Table 11.

Table 11: Empirical performance with multiple data-collection policies.

| Methods | GMV | ROI | Cost |
|---|---|---|---|
| **DiffBid** | 548.5 | 5.00 | 109.4 |
| **AIGB-Pearl (ours)** | 575.7 | 5.04 | 114.6 |
| $\Delta$ | +4.9% | +0.7% | +4.2% |

It can be observed that AIGB-Pearl still outperforms AIGB by 4.9%, indicating that its performance is robust to the offline dataset's specific distribution.

## F.8 HYPER-PARAMETER TUNING

We conduct sensitivity experiments with respect to the hyper-parameter $\delta_k$. For hyper-parameter $L_p$, we leverage the lower bound given in Eq. 10 in the main experiments. For the hyper-parameter $\epsilon$, we use the same empirical value of $5\%$ as in AIGB in the main experiments, which is typically set based on operational requirements.

Specifically, in the planner loss given in Eq. 12, $\beta_2$ is the penalty factor corresponding to the KL constraint. Actually, we control the KL divergence $\delta_k$ by tuning $\beta_2$. Table. 12 gives the hyper-parameter tuning results. It can be observed that, as long as $\delta_k$ is neither too large (in which case the algorithm degenerates into AIGB with a pure demonstration likelihood maximization term) nor too small (e.g., $\beta_2 = 0$, which completely removes the demonstration likelihood maximization), the proposed method consistently outperforms the original AIGB. This also validates the discussion on performance bounds in Appendix D, which demonstrates that a moderate $\delta_k$ balances the lower and upper bounds, thereby yielding optimal performance.

Table 12: Hyper-parameter tuning with respect to KL constraint $\delta_k$.

| $\beta_2$ Setting | 0 (no KL) | 0.1 | 0.5 | 1.0 | 2 | 5 | 10 | $\infty$ (AIGB, only KL) |
|---|---|---|---|---|---|---|---|---|
| $\delta_k$ values | 0.85 | 0.51 | 0.50 | 0.49 | 0.43 | 0.39 | 0.36 | 0.32 |
| GMV | 548.5 | 563.8 | 571.3 | **574.2** | 573.9 | 570.7 | 567.6 | 556.3 |

Table 13: Empirical performance in the simulated experiments with a first-price auction.

| Methods | GMV | ROI | Cost |
|---|---|---|---|
| **DiffBid** | 1,546 | 5.13 | 301 |
| **AIGB-Pearl (ours)** | 1,611 | 5.18 | 311 |
| **Δ** | **+4.2%** | **+1.0%** | **+3.3%** |

## G    EXTENDING AIGB-PEARL TO FIRST-PRICE AUCTIONS

We note that the proposed method remains effective in first-price auctions with a proper adaptation. Specifically, unlike second-price auctions where the optimal bid for impression $i$ takes the form $\text{bid}_i = av_i$, in first-price auctions, the optimal bid for impression $i$ is given by $\text{bid}_i = \min(av_i, p_i)$, which typically involves an extra bid shading method to predict the winning price (Gligorijevic et al., 2020; Wu et al., 2015). Equipped with an off-the-shelf bid shading method (whose design is beyond the scope of this work), the auto-bidding problem in a first-price auction remains an offline sequential decision problem, i.e., making decisions over an $a$-sequence, to which the proposed method applies directly.

To validate the effectiveness of the proposed method under first-price auctions, we additionally conduct a simulated first-price auction experiment against the state-of-the-art AIGB method, where all methods are equipped with the same bid shading method. The results are presented in Table 13, demonstrating the effectiveness of our proposed method.

## H    EXTENDING AIGB-PEARL TO ONLINE SETTING DISCUSSION

We note that AIGB-Pearl can be extended to online settings when equipped with a safe online exploration policy. Specifically, due to safety constraints, the auto-bidding policy during training cannot interact directly with the live advertising system; only safe exploration policies are permitted to collect data online (Mou et al., 2022). Consequently, an online auto-bidding framework typically involves two parts:

- a safe online exploration policy, which is a well-established component in existing work (Mou et al., 2022) and beyond the scope of this paper;
- an offline policy training method that leverages the data collected online.

AIGB-Pearl can be directly applied as the offline policy training method within the online framework without modification. In practice, due to the safety and stability concerns, many industrial auto-bidding systems adopt an offline optimization paradigm. For this practical reason, we focus on the offline setting in this work.

## I    LLM USAGE

The authors have used Large Language Models (LLMs) exclusively for grammar checking and lexical refinement during the writing process. No LLM-generated content, data analysis, or substantive contributions to the research methodology, results, or conclusions are involved in this work.

ACKNOWLEDGMENTS

This work was supported by Alibaba Group through the Alibaba Innovative Research Program.

