# OpenReview forum: "Enhancing Generative Auto-bidding with Offline Reward Evaluation and Policy Search"
_ICLR.cc/2026/Conference — ICLR 2026 Oral_

### Official Review · Reviewer_29sV · 2025-10-28

**Soundness:** 2
**Presentation:** 2
**Contribution:** 2
**Rating:** 4
**Confidence:** 3

**Summary:**

This paper studies how to integrate an offline reinforcement learning (offline RL) approach and conditional generative planning for the auto-bidding task of commercial advertisement. The proposed approach is akin to a conservative model-based RL method, where the evaluator is first trained in a pessimistic way with a Lipschitz constraint and then, controller is optimized to maximize the evaluator's score under the KL and Lipschitz constraints. The theoretical analysis is provided to bound the sub-optimality, and the experiment demonstrates that the proposed method works better than existing offline RL baselines and generative planning baselines.

**Strengths:**

- The motivation for combining conditional generative planning and optimization (score maximization) is well-explained.

- The proposed approach appears reasonable given the motivation of maximizing the score as an RL problem. In experiments, the proposed method is compared to offline RL baselines and works better than the compared method.

- A good ablation study is provided. Especially, qualitative analysis showing how the trajectory of the proposed method differs between w/ and w/o KL explains why Lipschitz constraints are not sufficient even under the Lipschitz condition, which answers one of the questions I had.

**Weaknesses:**

- I wonder where the conditional generative planning component is in the proposed method. While the proposed method works better than both traditional offline RL and generative planning-based baselines, given that the proposed method does not try to maximize the likelihood of demonstration, this looks more like a conservative model-based RL rather than a combination of planning and RL. The phrasing should be reconsidered, and in this sense, I wonder how impactful this paper is to the ICLR community (I acknowledge the contributions for the auto-bidding application, while in the offline RL context, Lipschitz constraints are another simple way to introduce conservativeness differently).

- Theorem 2 seems to lack a necessary assumption. In my understanding, the evaluator's bias cannot be measured without having an assumption of the evaluator's accuracy or how the evaluator is trained (e.g., the evaluator is trained to satisfy the Lipschitz condition). It seems that he necessary assumption is not provided in advance.

- Similarly, the paper claims that Theorem 1 is satisfied by "the evaluator's design" (i.e., by having $\root{T} R_m$ constraints in the regression). However, if this is the reason, the theorem should be about the Lipschitz continuity of $\hat{y}(\tau)$, not that of $y(\tau)$. There is no evidence presented regarding whether $y(\tau)$ satisfies the Lipschitz continuity. I believe this should be an "assumption", instead of a "theorem". Overall, the theoretical analysis does not seem rigorous, and these details should be carefully reviewed.

**Questions:**

- What is the sensitivity of the model like regarding the constraints (i.e., $\delta_K$ and $L_p$)? How are these values determined in the experiment, and how much effort is needed to adjust these hyperparameters?

---

> ### Author Response · Authors · 2025-11-21
> **Official Comment by Authors (1/3)**
>
> We sincerely thank Reviewer 29sV for these insightful and valuable comments. The remainders focus on concerns to address.
>
> ---
>
> **1.1 Conditional Generative Planning Component Clarification. (W1, part 1)**
>
> Thank you for the valuable comment. We note that in this work, we focus on improving the conditional generative **planner** $p_\theta(\tau|y)$ through the proposed score maximization method, which explicitly **includes** maximizing the likelihood of demonstrations from the offline dataset as one of its key objectives. (Note that the planner acts as the core of planning-and-control generative auto-bidding methods [1]. For the controller, we adopt the same off-the-shelf inverse dynamics model as used in AIGB, which is not the focus of this work.)
>
> **Generative Planner Model:** Specifically, the planner $p_\theta(\tau|y)$ takes a high-level goal (i.e., a target return $y$) as input and autoregressively generates the **entire** future state trajectory $\tau$, as implemented by the Causal Transformer shown in Figure 1 of the manuscript. This modeling fundamentally differs from the policy $\pi(a|s)$ in traditional RL, which typically predicts only a single-step action.
>
> **Generative Planner Training:** In our framework, the generative planner $p_\theta(\tau|y)$ is optimized via the proposed score-maximization objective formulated in Eq. (8) and implemented in Eq. (12) of the manuscript:
>
> $\text{the planner loss } l\_p(\theta)=\underbrace{-\mathbb{E}\_{\tau\sim p\_\theta(\tau|y^*)}[\hat{y}\_\phi(\tau)]}\_{\text{score maximization via RL}}-\underbrace{\beta\_2\mathbb{E}\_{(\tau,y)\sim p\_D}[\log p\_\theta(\tau|y)]}\_{\text{demonstration likelihood maximization}}+\text{Lipschitz penalty}$, (Eq.(12) in the original manuscript)
>
> where the second term maximizes the likelihood of demonstrations from the offline dataset, consistent with traditional generative planning-based methods (e.g., AIGB). Note that this term serves a dual role: (i) it makes the planner learns the conditional trajectory distribution in the dataset, and (ii) it acts as a regularizer that mitigates the OOD risk during the planner's score maximization (as required by the KL constraint Eq.(8a) in the manuscript).
>
> Notably, unlike the standard generative auto-bidding planning framework AIGB that relies solely on demonstration likelihood maximization, we take a further step: building upon the planner’s ability to fit the data conditional distribution, we enable it to autonomously explore higher-quality trajectories by maximizing an evaluator score. In this way, our method seamlessly integrates generative planning with RL.
>
> **1.2. Our Work Impact. (W1, part 2)**
>
> Thanks for the insightful comment. Our work falls in the scope of ICLR in application science, and our contribution lies in a theoretically grounded and practically effective RL-enhanced generative planning pipeline for auto-bidding, a critical real-world application that powers multi-billion-dollar digital advertising ecosystems nowadays (e.g., Google Ads, Meta Ads, Alibaba Taobao). More broadly, our approach can be adapted to other planning-intensive domains with similar offline settings (e.g., autonomous driving). In this sense, our work offers theoretical support and a practical blueprint for advancing the synergy between planning and RL.

---

> ### Author Response · Authors · 2025-11-21
> **Official Comment by Authors (2/3)**
>
> In the following, we first clarify weakness 3 and then weakness 2 for better clarity.
>
> ---
>
> **3. Relation Between Theorem 1 and Evaluator Design Clarification. (W3)**
>
> Thank you for the valuable comment. We kindly clarify that we do not claim that "Theorem 1 is satisfied by the evaluator's design". On the contrary, Theorem 1 is a property of the true trajectory quality function $y(\tau)$, which holds due to the structure of the considered auto-bidding problem itself (as proved in Appendix C.1 in the original manuscript). Theorem 1 is **independent** of any model or evaluator design.
>
> We clarify that the evaluator's Lipschitz design is **motivated by** Theorem 1, rather than the basis for it. Specifically, in the original manuscript, line 193 states that:
>
> > Based on this property, we incorporate a $\sqrt{T}R_m$-Lipschitz continuity constraint into the evaluator’s design (as described in Section 3.2.1).
>
> Here, "this property" refers to Theorem 1. Since the true trajectory quality function $y(\tau)$ is $\sqrt{T}R_m$-Lipschitz, we regularize the evaluator $\hat{y}_\phi(\tau)$ to enforce the same smoothness property, ensuring that the trained evaluator inherits the Lipschitz continuity of the true trajectory quality $y(\tau)$. Since the Lipschitz constant of the trained evaluator may not be exactly $\sqrt{T}R_m$, we denote the trained evaluator's Lipschitz constant as $k\sqrt{T}R_m$, where $k$ represents the violation degree.
>
> We apologize for any confusion caused by the original presentation. We have further strengthened the clarity of line 193 in the revised manuscript as follows:
>
> > Motivated by Theorem 1, we enforce a $\sqrt{T}R_m$-Lipschitz regularity on the evaluator's training to inherit the Lipschitz continuity of the true trajectory quality $y(\tau)$ (as described in Section 3.2.1). Since the Lipschitz constant of the trained evaluator may not be exactly $\sqrt{T}R_m$, we denote it as $k\sqrt{T}R_m$, where $k\ge 0$ reflects the violation degree.
>
> ---
>
> **2. Bias Assumption & Training Procedure Clarification. (W2)**
>
> Thanks for the valuable comment. We note that the necessary accuracy assumption of the evaluator and its training procedure are included in the original manuscript. Specifically, the evaluator is trained to (i) fit the ground-truth trajectory qualities $y(\tau)$ on the offline dataset via an MSE loss, and (ii) satisfy the $\sqrt{T}R_m$-Lipschitz condition. This design is first mentioned in Section 3.1 (line 163 and line 194) in the original manuscript, and then detailed in Section 3.2.1 (line 277-286) of the original manuscript, where Eq. (11) specifies the evaluator’s training loss as:
>
> > Eq. (11): the evaluator loss $l_e$ = MSE loss + Lipschitz penalty.
>
>
> Moreover, the necessary assumptions regarding the evaluator accuracy on the training data and its Lipschitz constant used in Theorem 2 are provided in advance before giving the theoretical results in Theorem 2 in the original manuscript:
>
> > line 200 (the first sentence in Theorem 2) defines $\delta_D$ as the upper bound on the evaluator’s bias over the training dataset.
>
> > line 195 denotes the trained evaluator's Lipschitz constant as $k\sqrt{T}R_m$.
>
> Here, $\delta_D$ can be made small through the MSE loss, and k can be driven closer to 1 via the Lipschitz penalty. Theorem 2 is derived precisely based on these two pre-specified and controllable quantities.
>
> We apologize for any confusion caused by the original presentation, and we have further strengthened the clarity of this exposition in the revised manuscript as follows:
>
> > We have added an explicit MSE loss at line 163 when first introducing the evaluator to clarify that its basic training objective is MSE.
>
> > We have modified line 194 to "Motivated by Theorem 1, we enforce a $\sqrt{T}R_m$-Lipschitz regularity on the evaluator's training to inherit the Lipschitz continuity of the true trajectory quality $y(\tau)$" to make the logic clear.
>
> > We have added a further explanation on $\delta_D$ in the first sentence in Theorem 2, i.e., "$\mathbb{E}\_{\tau\sim D}|y(\tau)-\hat{y}_\phi(\tau)|\le \delta_D$" to make the meaning of $\delta_D$ more clearer.

---

> ### Author Response · Authors · 2025-11-21
> **Official Comment by Authors (3/3)**
>
> **4. Hyper-parameter Sensitivity & Determination. (Q1)**
>
> Thank you for the valuable question.
>
> **Determination of $L_p$:** We clarify that Theorem 3 indicates that the sub-optimality gap of our method is proportional to $L_p$, and we explicitly give the lower bound of $L_p$ in Eq. (10) of the original manuscript, i.e.,
>
> $L_p\ge \sup_{y_1\neq y_2}\frac{W_1(p_D(\tau|y_1),p_D(\tau|y_2))}{|y_1-y_2|}$.
>
> Therefore, we determine $L_p$ as its lower bound in the experiments and do not need to adjust it. As described in Section 4.4 in the original manuscript, we set $L_p=0.50$, close to its lower bound estimation in the experiment. (Note that, considering that the lower bound estimation is computed solely in a data-driven manner and may be underestimated, we conservatively increase its value accordingly.)
>
>
> **Sensitivity & Determination of $\delta_K$:** Note that in the practical algorithm design presented in Eq. (12) of the original manuscript, we employ demonstration likelihood maximization to ensure that the planner $p_\theta(\tau|y)$ remains close to the conditional trajectory distribution of the offline dataset $p_D(\tau|y)$, as required by the KL constraint.
>
> $\text{The planner loss } l\_p(\theta)=\underbrace{-\mathbb{E}\_{\tau\sim p\_\theta(\tau|y^*)}[\hat{y}\_\phi(\tau)]}\_{\text{score maximization via RL}}-\underbrace{\beta_2\mathbb{E}\_{(\tau,y)\sim p\_D}[\log p\_\theta(\tau|y)]}\_{\text{demonstration likelihood maximization}}+\text{Lipschitz Penalty}$
>
> Accordingly, in our experiments, we control the effective KL tolerance $\delta_K$ by tuning the hyperparameter $\beta_2$: a larger $\beta_2$ corresponds to a tighter (i.e., smaller) $\delta_K$. We conduct extra simulated experiments to examine the sensitivity of the model with respect to $y$, and the results are given below:
>
> | $\beta_2$ setting | 0 (no demonstration likelihood maximization) | 0.1 |0.5| 1(our choice) |2|5| 10 | approximate $\infty$ (only demonstration likelihood maximization, AIGB)|
> |---|---|---|---|---|---|---|---|---|
> |  $\delta_K$  value | 0.85  |  0.51 |0.50|0.49|0.43|0.39|0.36|0.32|
> |GMV| 548.5  | 563.8  |571.3|**574.2**|573.9|570.7|567.6|556.3|
>
>
> It can be observed that as long as $\beta_2$ is neither too large, where the algorithm would degenerate into AIGB with pure demonstration likelihood maximization term, nor too small (e.g.,  $\beta_2=0$, which completely removes the demonstration likelihood maximization), the proposed method consistently outperforms the original AIGB.
>
> In the experiments reported in the original manuscript, we set $\beta_2=1$, which yields the highest GMV performance among all tested configurations above. This setting achieves a 3.2% (=(574.2−556.3)/556.3) GMV uplift compared to the AIGB with pure demonstration likelihood maximization term (where $\beta_2\rightarrow\infty$).
>
> Notably, in the context of auto-bidding systems, a +3.2% GMV uplift is substantial, which translates to millions of RMB in additional daily GMV at Taobao-scale advertising platforms.
>
> ---
>
> **References:**
>
> [1] Jiayan Guo, et al. (2024). Generative Auto-bidding via Conditional Diffusion Modeling. KDD. 5038–5049.
>
>
> ---
>
> *Finally, thanks again for your time and helpful comments. We hope your concerns are well addressed. For any other questions, we are happy to discuss them and make further clarifications.*

---

### Official Review · Reviewer_6mYT · 2025-10-30

**Soundness:** 4
**Presentation:** 3
**Contribution:** 3
**Rating:** 8
**Confidence:** 2

**Summary:**

The paper proposes AIGB-Pearl (Planning with EvaluAtor via RL), a method that augments generative auto-bidding (AIGB) with a learned trajectory evaluator and a KL–Lipschitz constrained score-maximization procedure to enable safe exploration beyond an offline dataset. The work combines theoretical analysis (evaluator bias bounds and a sub-optimality gap under KL/Lipschitz constraints), a practical planner/evaluator architecture (synchronous coupling, Lipschitz regularization), and extensive simulated and real-world A/B experiments that report consistent GMV/ROI improvements over strong baselines.

**Strengths:**

1. Integrating an evaluator into a generative planner and constraining planning by KL + Lipschitz criteria to control evaluator bias is a principled and timely idea for offline auto-bidding.
2. The paper gives provable bounds (evaluator bias → performance gap; sub-optimality bound under the proposed constraints) that make the method more convincing.
3. Results include both simulated and large real-world A/B tests (TaoBao), showing consistent gains over state-of-the-art baselines and sensible ablations.
4. The synchronous coupling and the proposed surrogate for the Wasserstein term are well-motivated and seem implementable in practice.

**Weaknesses:**

1. In the problem statement the intrinsic impression value is introduced as v_i>0 (Section 2 / preliminaries). Later, in the experimental settings (Tables 5/6) the “value of impressions” is reported as ranging from 0 to 1. This inconsistency leaves readers unclear whether the theory (which uses v_i>0 and constants like R_m = max_i v_i/p_i) assumes arbitrary positive values or implicitly assumes normalized values in [0,1]. It also affects interpretability of constants such as R_m` Lipschitz constants that depend on reward magnitudes, and the hyperparameter choices.
2. In the simulated-experiment paragraph the manuscript text says: “The maximum and minimum budgets of advertisers are 1000 Yuan and 4000 Yuan, respectively.” This phrasing is inverted relative to Table 5, which lists Minimum budget = 1000 and Maximum budget = 4000. In the real-world experiment paragraph a similar inversion appears: “The maximum and minimum budgets of advertisers are 50 Yuan and 10,000 Yuan, respectively.” This contradicts Table 6 (Minimum = 50, Maximum = 10,000). Such directional swaps are minor editorial errors but can confuse reproducibility and readers’ understanding of budget regimes used in experiments. Given that the budget ranges are central to the task formulation, they should be unambiguous.

**Questions:**

See weakness

---

> ### Author Response · Authors · 2025-11-21
>
> We sincerely thank Reviewer 6mYT for these valuable comments. The remainders focus on concerns to address.
>
> ---
>
> **1. The Range of $v_i$. (W1)**
>
> Thanks for the valuable comment. We note that our theoretical results and proposed algorithms are developed under arbitrary positive impression values $v_i>0$, without imposing any specific upper bound. All the values introduced in the paper, including $R_m$ and the Lipschitz value, are defined for general positive values of $v_i$. The precise meaning, and thus the range of $v_i$, depends on the specific optimization objective set by the advertiser:
>
> - If the advertiser's objective is to maximize the number of clicks or conversions, then $v_i$ corresponds to the click-through rate (CTR) or the conversion rate (CTCVR), respectively, both in the range of $[0,1]$;
> - If the advertiser's objective is to maximize the GMV, then $v_i$ corresponds to the GMV of the impression, which can be unbounded.
>
> In our experiments, we consider the GMV objective, and we have corrected the typo from $v_i\in [0,1]$ to $v_i>0$ in Table 5 and Table 6 of the revised manuscript.
>
> ---
>
> **2. Budget Range Correction. (W2)**
>
> Thanks for pointing out the inconsistency. We correct the textual descriptions to ensure they are consistent with the tables in the revised manuscript, which now reads as:
>
> > The minimum and maximum budgets of advertisers are 1000 Yuan and 4000 Yuan, respectively.
> > The minimum and maximum budgets of advertisers are 50 Yuan and 10,000 Yuan, respectively.
>
>
>
> ---
>
> *Finally, thanks again for your time and helpful comments. We hope your concerns are well addressed. For any other questions, we are happy to discuss them and make further clarifications.*

---

### Official Review · Reviewer_wJct · 2025-10-30

**Soundness:** 3
**Presentation:** 3
**Contribution:** 2
**Rating:** 6
**Confidence:** 3

**Summary:**

This paper proposes a novel method (AIGB-Pearl) to enhance generative auto-bidding (AIGB) by integrating
reinforcement learning (RL) into a trajectory-based generative modeling framework. The key idea is to
train a trajectory evaluator to score the quality of generated trajectories and use this score to guide the generative
planner to find higher-performance trajectories beyond the offline dataset, while ensuring safety and
generalization via KL and Lipschitz constraints. The method successfully leads to more stable training than
traditional offline RL, and is supported by theoretical guarantees on sub-optimality and generalization. The
authors conduct extensive experiments in simulated and real-world advertising environments to demonstrate
state-of-the-art performance.

**Strengths:**

1. Originality: The integration of RL into a generative auto-bidding framework is innovative and addresses
the inability to explore beyond the offline dataset of existing AIGB methods. The introduction of a
trajectory evaluator and the use of KL-Lipschitz constraints to ensure safe exploration are also novel
contributions.

2. Theoretical Rigor: The paper provides a thorough theoretical analysis, including bounds on evaluator
bias, sub-optimality and generalization. The use of Wasserstein distance and synchronous coupling for
Lipschitz regularization is well-motivated and technically sound.

3. Significance: The method is deployed in a real-world advertising platform and shows significant improvements
compared to previous methods, indicating high practical relevance.

4. Ablation Analysis: Ablation studies clearly demonstrate the importance of both KL and Lipschitz
constraints. The analysis enhances the rigor and completeness of the article’s theoretical framework.

5. Experimental Validation: Comprehensive and detailed experiments are conducted in both simulated
and real-world environments with large-scale datasets and multiple budget levels, demonstrating the
superior performance of the algorithm.

**Weaknesses:**

Theoretical Assumptions: Some theoretical results in this paper rely on strong Assumption 1, which can be hard to verify on real-world dataset.

**Questions:**

1. The paper considers the second-price bidding problem. How would the core formulation need to be adapted
for a first-price auction environment? Will the method still be effective under first-price settings?

2. The theoretical analysis assumes the offline dataset satisfies LSI. What is the performance of AIGBPearl
if the assumption does not hold? Are there any ways to relax the assumption?

3. The paper considers offline RL baselines. Can the framework be extended to online cases?

---

> ### Author Response · Authors · 2025-11-21
> **Official Comment by Authors (1/2)**
>
> We sincerely thank Reviewer wJct for these valuable comments. The remainders focus on concerns to address.
>
> ---
>
> **1. Reasonableness of LSI Assumption. (W1)**
>
> Thank you for the valuable comments. We explain the reasonableness of the LSI assumption in the considered problem. Note that in many real-world auto-bidding systems, including the one considered in the paper, due to safety operational constraints, the online-deployed bidding policy is typically a single fixed model, and the offline dataset is collected exclusively from this single policy over multiple days. For example, in the considered auto-bidding system, the online-deployed baseline policy is a conditional generative model that generates identical trajectory plans for a given advertiser under identical conditions each day. The variation across different trajectories of the same advertiser in the offline dataset is solely due to stochastic environmental factors (e.g., traffic fluctuations). Since these exogenous perturbations are typically the sum of many independent impression-level sources of noise, the resulting trajectory deviations can be reasonably approximated as a Gaussian distribution, which is known to satisfy the LSI [1] Assumption.
>
> ---
>
> **2. LSI Assumption Relaxation. (Q2)**
>
> Thank you for the insightful comments. We conduct additional experiments in settings where Assumption 1 does not hold, and the results show that AIGB-Pearl still outperforms the state-of-the-art AIGB method. Through further theoretical analysis, we demonstrate that Assumption 1 **is not necessary** for establishing the proposed theoretical guarantees. Accordingly, we have **removed** it in the revised manuscript, with the analysis now grounded in fundamental inequalities. We also note that our framework remains fully valid in the absence of Assumption 1, demonstrating its generality. Specifically:
>
> **(1) Empirical Performance When Assumption 1 Does Not Hold.**
>
> We collect trajectories in the simulated environment using nine distinct bidding policies, thereby constructing an offline dataset with a multi-modal distribution that violates the LSI. The empirical results are given below.
>
> | Methods |  GMV | ROI | Cost |
> | -------- | -------- | -------- |---|
> | AIGB     |  548.5    | 5.00    |109.4
> |AIGB-Pearl (ours) | 575.7|5.04|114.6|
> |diff|**+4.9%** |**+0.7%**|**+4.2%**|
>
> We observe that AIGB-Pearl still outperforms AIGB by +4.9%, indicating that its performance is robust to the validity of Assumption 1. We therefore proceed to theoretically analyze the necessity of Assumption 1 and find that it is not necessary for establishing the proposed theoretical guarantees.
>
> **(2) Assumption 1 Is Not Necessary: A Theoretical Justification.**
>
> Note that Assumption 1 is used only to ensure that the Wasserstein distance term $\mathbb{E}\_{\tau\sim p_D(\tau|y)}[W_1(p_\theta(\tau|y), p_D(\tau|y))]$ can be bounded by the KL-divergence in the original theoretical derivation. Nonetheless, we find that this property holds naturally in the considered auto-bidding problem without the need for Assumption 1. Specifically, we show that the trajectory space has a finite diameter, which enables us to bound the Wasserstein distance in terms of the KL divergence using only fundamental information-theoretic inequalities (Pinsker and Wasserstein-TV inequalities). This yields a more practical and assumption-free analysis while preserving the core theoretical result.
>
> A proof sketch is as follows:
>
> > With Pinsker inequality [2] and the bounded-diameter transport inequality [3], we have:
>
> > $W_1(p_\theta(\tau|y), p_D(\tau|y))\le \text{diam}(\mathcal{T})\times TV(p_\theta(\tau|y), p_D(\tau|y))\le \text{diam}(\mathcal{T})\times\sqrt{\frac{1}{2}KL(p_D(\tau|y)\| p_\theta(\tau|y))},$
>
> > where $\text{diam}(\mathcal{T})$ denotes the diameter of the trajectory space$^*$. The trajectory space is:
>
> > $\mathcal{T}=\lbrace[[1, \bar{c}\_{0},x], [2,\bar{c}\_{1}, x], \cdots,[T, \bar{c}\_{T-1}, x]]| \bar{c}\_t\ge 0, \forall t, \text{and} \sum\_t \bar{c}\_t\le 1\rbrace$.
>
> > We prove that the diameter of $\mathcal{T}$ can be bounded by $\sqrt{2}$.
>
> > In this way, controlling the KL-divergence, i.e., $\mathbb{E}\_{\tau\sim p_D(\tau|y)}[KL(p_D(\tau|y)\| p_\theta(\tau|y))]\le \delta_K$, can bound the Wasserstein distance:
>
> > $\mathbb{E}\_{\tau\sim p_D(\tau|y)}[W_1(p_\theta(\tau|y), p_D(\tau|y))]\le \sqrt{\delta_K},$
>
> > where Assumption 1 is **not** used.
>
> We give the detailed proof in Appendix C.4 of the revised manuscript. We want to note that the removal of Assumption 1 does not alter the manuscript’s core theoretical analysis (only a minor adjustment on Eq. (7) and Eq. (9)), algorithmic framework, or experimental results.
>
> $^*$ *Note that we here normalize the cost in the trajectory by the budget $B$, i.e., $\bar{c}= c/B$, and the static advertiser-specific feature $x$ includes the budget information to ensure comparability across advertisers, which also better aligns with our feature design in practice.*

---

> ### Author Response · Authors · 2025-11-21
> **Official Comment by Authors (2/2)**
>
> **3. Adapting to First-Price Auctions. (Q1)**
>
> Thank you for the insightful question. We note that the proposed method remains effective in first-price auctions with a proper adaptation. Specifically, unlike second-price auctions where the optimal bid for impression $i$ takes the form $\text{bid}_i=av_i$, in first-price auctions, the optimal bid for impression $i$ is given by $\text{bid}_i=\min(av_i, p_i)$, which typically involves an extra bid shading method to predict the winning price $p_i$ [4,6]. Equipped with an off-the-shelf bid shading method (whose design is beyond the scope of this work), the auto-bidding problem in a first-price auction remains an offline sequential decision-problem, i.e., making decisions over an $a$-sequence, to which the proposed method applies directly.
>
> To validate the proposed method's effectiveness under first-price auctions, we additionally conduct a simulated first-price auction experiment against the state-of-the-art AIGB method, where all methods are equipped with the same bid shading method. The results are given below, which demonstrate our method's effectiveness.
>
> | Methods | GMV | ROI |Cost|
> | -------- | -------- | -----------| ---|
> | AIGB    |  1,546    | 5.13 |301|
> |AIGB-Pearl (ours)| 1,611|5.18|311|
> |diff|**+4.2%**|**+1.0%**|**+3.3%**|
>
> ---
>
> **4. Extension to Online Settings. (Q3)**
>
> Thank you for the valuable question. We note that AIGB-Pearl can be extended to online settings when equipped with a safe online exploration policy. Specifically, due to safety constraints, the auto-bidding policy **during training** cannot interact directly with the live advertising system; only safe exploration policies are permitted to collect data online [5]. Consequently, an online auto-bidding framework typically involves two parts:
>
> - a safe online exploration policy, which is a well-established component in existing work [5] and beyond the scope of this paper;
> - an offline policy training method leveraging the data collected online.
>
> AIGB-Pearl can directly act as the offline policy training method in the online framework without modifications. In practice, due to the safety and stability concerns, many industrial auto-bidding systems adopt an offline optimization paradigm. For this practical reason, we focus on the offline setting in this work.
>
> ---
>
> **References:**
>
> [1] Gross, L. (1975). Logarithmic Sobolev Inequalities. American Journal of Mathematics, 97(4), 1061–1083.
>
> [2] Alexandre B. (2008). Tsybakov. Introduction to Nonparametric Estimation. Springer Publishing Company, Incorporated.
>
> [3] Villani, Cédric. (2008). Optimal transport: old and new. Vol. 338. Berlin: Springer.
>
> [4] Gligorijevic, D., et al. (2020). Bid shading in the brave new world of first-price auctions. CIKM (pp. 2453-2460).
>
> [5] Mou Z., et al. (2022). Sustainable online reinforcement learning for auto-bidding. NeurIPS (pp. 2651–2663).
>
> [6] Wu, et al. (2015). Predicting winning price in real-time bidding with censored data. KDD (pp. 1305-1314).
>
> ---
>
> *We sincerely thank the reviewer again for your insightful and constructive comments, which prompted us to reflect more deeply on our work and, as a result, enabled us to improve the quality of our manuscript. Finally, we hope your concerns are well addressed. For any other questions, we are happy to discuss them and make further clarifications.*

---

### Official Review · Reviewer_3d4u · 2025-11-01

**Soundness:** 3
**Presentation:** 3
**Contribution:** 3
**Rating:** 6
**Confidence:** 3

**Summary:**

The paper introduces AIGB-Pearl, a novel method designed to enhance AI-Generated Bidding (AIGB) by overcoming its limitation of imitating trajectories from a static offline dataset. AIGB-Pearl integrates generative planning with policy optimization by first constructing a trajectory evaluator through supervised learning to provides the necessary reward signal to guide the generative model. To ensure reliable utilization of this evaluator and to mitigate the Out-of-Distribution (OOD) issue, the method designs and enforces a provably sound KL-Lipschitz-constrained score maximization objective for the generative planner. AIGB-Pearl achieves greater training stability and demonstrates SOTA performance in extensive simulated and real-world advertising systems, consistently achieving improvement in GMV and other metrics compared to prior SOTA AIGB method

**Strengths:**

1. Novelty. AIGB-Pearl is the first work that integrates generative planning and policy optimization to address the performance bottleneck of existing AIGB methods. The integration of offline RL methods into the AIGB is important to guide the policy training and exploration of new trajectories
2. Technical Contribution. The paper provides provable theoretical guarantees to ensure the reliability and safety of the policy optimization, which is critical when exploring beyond the static offline dataset. The core innovation is the design of a provably sound KL-Lipschitz-constrained score maximization objective that help mitigates the OOD problem
3. Evaluation. In both the simulation and the real-world online A/B test, the methods achieve SOTA performance in the auto-bidding problem, out-performing the strong benchmark AIGB.

**Weaknesses:**

1. The paper provides details on the overall dataset sizes: the simulated experiment uses 5k trajectories generated by 20 advertisers, while the real-world experiment uses 200k trajectories from 10k advertisers. However, it doesn’t explicitly state how many of these trajectories are used to train the evaluator model, and 5k/200k appears relatively small for training a stable evaluator capable of mitigating OOD issues.

2. The paper states and assumes that the offline data distribution must satisfy the Logarithmic Sobolev Inequality (LSI) with a positive constant ρ, and this assumption could be satisfied through strategic adjustment of the trajectory sampling probability per condition y in the offline dataset D, the need to potentially alter the raw offline dataset distribution to ensure the theoretical bounds hold represents a limiting constraint or assumption on the data

**Questions:**

1. The effectiveness of AIGB-Pearl hinges on the reliability of the trajectory evaluator. The theoretical analysis bounds the gap between the planner's score and its true performance, where one of the core terms is the evaluator's bias on its training dataset. Could the authors provide the measured value of the evaluator's bias in both the simulated and real-world experiments?

2. Related to weakness 2, Regarding “this assumption could be satisfied through strategic adjustment of the trajectory sampling probability per condition y in the offline dataset D”, Could the authors detail the specific nature and extent of this strategic adjustment applied to the offline dataset (e.g., which trajectories or conditions were adjusted)?

---

> ### Author Response · Authors · 2025-11-21
> **Official Comment by Authors (1/3)**
>
> We sincerely thank Reviewer 3d4u for the insightful comments. The remainders focus on concerns to address.
>
> ---
>
>
> **1. Evaluator Training Data & Capability of Mitigating OOD Issues. (W1)**
>
> Thank you for the valuable comments.
>
> **(1) Evaluator Training Data.**
>
> We clarify that the evaluator is trained on the entire offline dataset, i.e., 5k trajectories from 250 advertisers in the simulated experiments and 200k trajectories from 10k advertisers in the real-world experiments. This detail has been added to the Experiment Setup in Section 4.1 of the revised manuscript.
>
> We note that this data scale is frequently used in the auto-bidding literature [1,2]. It typically reflects the operational reality of the auto-bidding system, including the number of advertisers and the available data collection duration, rather than being an arbitrary design choice. Specifically, each advertiser contributes one trajectory per day in the considered auto-bidding system. Collecting a large number of trajectories from a fixed set of advertisers requires a multi-day aggregation process. In our experiments, the real-world dataset consists of 200k trajectories from 10k advertisers, collected over 20 days; similarly, the simulated dataset contains 5k trajectories from 20 advertisers, equivalent to 250 simulation days.
>
> We acknowledge that a larger dataset may further enhance the evaluator’s accuracy. Nonetheless, our work primarily focuses on improving model performance with a given dataset. Exploring ways to boost performance through expanded data collection further is a promising direction for future work.
>
> **(2) Evaluator Accuracy on OOD Data.**
>
> We examine the generalization ability of the trained evaluator using $K$-fold cross-validation ($K=5$). Specifically, we evaluate its accuracy on the OOD test data in cross-validation along two dimensions, including
> - the *absolute accuracy* by mean absolute error (MAE) metric, reflecting how close the predicted scores are to ground truth scores, and
> - the *ranking accuracy* by AUC metric, reflecting the correctness of relative rankings between trajectory pairs,
>
> where the MAE is normalized by the advertiser's budget to ensure comparability across advertisers. The results are presented below and have been added to Section 4.5 of the revised manuscript.
>
> |   Simulated Experiments  | Training Data | OOD Data (in Cross-Validation) |  Real-world Experiments  | Training Data |OOD Data (in Cross-Validation) |
> |----------|-----------|------------|----------|-----------|------------|
> |MAE $\downarrow$|0.6|0.7 $\pm$ 0.06|MAE $\downarrow$|1.0|1.2 $\pm$ 0.03|
> |AUC  $\uparrow$ |  89.9%|85.5% $\pm$ 0.5%|AUC  $\uparrow$ | 77.4% |75.1% $\pm$ 0.2% |
>
> To the best of our knowledge, we are the **first** to introduce the trajectory evaluator into the generative auto-bidding framework. The reasonableness of our evaluator can be evidenced by its pairwise ranking accuracy of 86% and 75% AUC on OOD trajectories in the simulated and real-world experiments, respectively, which are substantially above the 50% random chance level, despite the high complexity and dynamic nature of the bidding environment. Moreover, the **small gap** between training and OOD test performance (i.e., a 2% to 4% drop in AUC) demonstrates the evaluator's ability to generalize under distributional shifts. Importantly, with the guidance of the trained evaluator, the planner outperforms state-of-the-art AIGB methods even on OOD data, as described below.
>
> **(3) Planner Performance on OOD Data.**
>
> We note that the planner’s performance on unseen advertisers is reported in Table 3 of the original manuscript. The results show that AIGB-Pearl consistently outperforms state-of-the-art AIGB methods by **+3%** in GMV, demonstrating its strong generalization capability under the guidance of the evaluator. This validates the generalization ability of the trained evaluator. (It is also worth noting that, in the context of auto-bidding systems, a +3\% GMV uplift is highly significant, translating to millions of RMB in additional daily GMV on Taobao-scale advertising platforms.)
>
> ---
>
> **2. Evaluator Bias on Training Data. (Q1)**
>
> Thank you for the valuable question. In the table above in our response 1, we give the evaluator's bias $\delta_D$ on the training data that is used in the performance-gap theorem. Specifically, $\delta_D$ corresponds to the evaluator's MAE on the training data, which is 0.6 in the simulated experiments and 1.0 in the real-world experiments, respectively. (Note that the trajectory quality ground truth here is normalized by the budget, with an absolute value of around 6.) Importantly, as given in Tables 1 and 2 in the original manuscript, the planner outperforms state-of-the-art AIGB methods by +3% in GMV under the guidance of the trained evaluator.

---

> ### Author Response · Authors · 2025-11-21
> **Official Comment by Authors (2/3)**
>
> **3. Assumption 1 and Strategic Adjustment Clarification. (W2&Q2)**
>
> Thank you for the insightful comments. In the following, we first explain the reasonableness of Assumption 1 in the context of the considered auto-bidding problem and detail both the nature and extent of the strategic adjustment. Then, through further theoretical and empirical analysis, we demonstrate that Assumption 1 **is not necessary** for establishing the proposed theoretical guarantees. Accordingly, we have **removed** it in the revised manuscript, with the analysis now grounded in fundamental inequalities. We also note that our framework remains fully valid in the absence of Assumption 1, demonstrating its generality. Specifically:
>
> **(1) Reasonableness of Assumption 1.**
>
> We note that in many real-world auto-bidding systems, including the considered one, due to safety operational constraints, the online-deployed bidding policy is typically a single fixed model, and the offline dataset is collected exclusively from this single policy over multiple days, where an advertiser contributes a single trajectory per day. For example, in the considered auto-bidding system, the online-deployed baseline policy is a conditional generative model that generates identical trajectory plans for a given advertiser under identical conditions each day. The variation across different trajectories of the same advertiser in the offline dataset is solely due to stochastic environmental factors (e.g., traffic fluctuations). Since these exogenous perturbations are typically the sum of many independent impression-level sources of noise, the resulting trajectory deviations can be reasonably approximated as a Gaussian distribution, which is known to satisfy the LSI [3] as required in Assumption 1.
>
> **(2) Nature and Extent of The Strategic Adjustment.**
>
> We note that although the offline dataset $p_D(\tau|y)$ ideally exhibits approximately Gaussian-like behavior, as explained above, real-world data may contain outliers resulting from atypical advertiser behaviors. For example, advertisers sometimes stop campaigns early or frequently change bidding settings. These actions may create irregular or sparse trajectory patterns, which break the smooth, light-tailed structure needed for the LSI assumption. The “strategic adjustment” mentioned in the original manuscript refers to the removal of **outliers**, which is inherently a data cleaning process, and the corresponding optimization formulation is intended to illustrate the theoretical approach underlying it. Specifically, we remove any trajectory in the offline dataset that meets one or more of the following outlier conditions:
> - The trajectory contains fewer than $0.6T$ online bidding time steps.
> - The trajectory involves a budget change of more than 20% by the advertiser during the bidding episode.
>
> More precisely, we here clarify that the strategic adjustment does not alter the essential distribution of the offline dataset; it merely removes approximately 5% of anomalous trajectories.
>
> **(3) Empirical Performance When Assumption 1 Does Not Hold.**
>
> We further examine the performance of the proposed method when Assumption 1 is not met. Specifically, we collect trajectories in the simulated environment using nine distinct bidding policies, thereby constructing an offline dataset with a multi-modal distribution that violates the LSI. The empirical results are given below.
>
> | Methods |  GMV | ROI | Cost |
> | -------- | -------- | -------- |---|
> | AIGB     |  548.5    | 5.00    |109.4
> |AIGB-Pearl (ours) | 575.7|5.04|114.6|
> |diff|**+4.9%** |**+0.7%**|**+4.2%**|
>
> We observe that AIGB-Pearl still outperforms AIGB by +4.9%, indicating that its performance is robust to the validity of Assumption 1. We therefore proceed to analyze the theoretical necessity of Assumption 1 as follows.

---

> ### Author Response · Authors · 2025-11-21
> **Official Comment by Authors (3/3)**
>
> **(4) Assumption 1 Is Not Necessary: A Theoretical Justification.**
>
> We next theoretically demonstrate that Assumption 1 is not necessary for establishing the proposed theoretical guarantees and can be removed.
>
> Note that Assumption 1 is used only to ensure that the Wasserstein distance term $\mathbb{E}\_{\tau\sim p_D(\tau|y)}[W_1(p_\theta(\tau|y), p_D(\tau|y))]$ can be bounded by the KL-divergence in the original theoretical derivation. Nonetheless, we find that this property holds naturally in the considered auto-bidding problem without the need for Assumption 1. Specifically, we show that the trajectory space has a finite diameter, which enables us to bound the Wasserstein distance in terms of the KL divergence using only fundamental information-theoretic inequalities (Pinsker inequality and Wasserstein-TV inequality). This yields a more practical and assumption-free analysis while preserving the core theoretical result.
>
> A proof sketch is as follows:
>
>
> > With Pinsker inequality [4] and the bounded-diameter transport inequality [5], we have:
>
> > $W_1(p_\theta(\tau|y), p_D(\tau|y))\le\text{diam}(\mathcal{T})\times TV(p_\theta(\tau|y), p_D(\tau|y))\le \text{diam}(\mathcal{T})\times\sqrt{\frac{1}{2}KL(p_D(\tau|y)\| p_\theta(\tau|y))},$
>
> > where $\text{diam}(\mathcal{T})$ denotes the diameter of the trajectory space$^*$. The trajectory space is:
>
> >$\mathcal{T}=\lbrace [[1, \bar{c}_{0},x], [2,\bar{c}\_{1}, x], \cdots,[T, \bar{c}\_{T-1}, x]]| \bar{c}\_t\ge 0, \forall t, \text{and} \sum\_t \bar{c}\_t\le 1 \rbrace$
>
> > We prove that the diameter of $\mathcal{T}$ is bounded by $\sqrt{2}$.
>
> > In this way, controlling the KL-divergence, i.e., $\mathbb{E}\_{\tau\sim p\_D(\tau|y)}[KL(p\_D(\tau|y)\| p\_\theta(\tau|y))]\le \delta\_K$, can bound this Wasserstein distance:
>
> > $\mathbb{E}\_{\tau\sim p_D(\tau|y)}[W_1(p_\theta(\tau|y), p_D(\tau|y))]\le \sqrt{\delta_K},$
>
> > where Assumption 1 is **not** used.
>
>
> We give the detailed proof in Appendix C.4 of the revised manuscript and remove Assumption 1 in the main content. We would like to note that the removal of Assumption 1 does not alter the manuscript’s theoretical analysis (only a minor adjustment on Eq. (7) and Eq. (9)), algorithmic framework, or experimental results.
>
>
> $^*$ *Note that we here normalize the cost in the trajectory by the budget constant $B$, i.e., $\bar{c}_t= c_t/B$, and the static advertiser-specific feature $x$ includes the budget information to ensure comparability across advertisers, which also better aligns with the feature design in practice.*
>
> ---
>
> **References:**
>
> [1] Jiayan Guo, et al. (2024). Generative Auto-bidding via Conditional Diffusion Modeling. KDD. 5038–5049.
>
> [2] Yewen Li, et al. (2025). GAS: Generative Auto-bidding with Post-training Search. WWW. 315–324.
>
> [3] Gross, L. (1975). Logarithmic Sobolev Inequalities. American Journal of Mathematics, 97(4), 1061–1083.
>
> [4] Alexandre B. (2008). Tsybakov. Introduction to Nonparametric Estimation. Springer Publishing Company, Incorporated.
>
> [5] Villani, Cédric. (2008). Optimal transport: old and new. Vol. 338. Berlin: Springer.
>
> ---
>
> *We sincerely thank the reviewer again for your insightful and constructive comments, which prompted us to reflect more deeply on our work and, as a result, enabled us to improve the quality of our manuscript. Finally, we hope your concerns are well addressed. For any other questions, we are happy to discuss them and make further clarifications.*

---

> > ### Comment · Reviewer_3d4u · 2025-11-28
> >
> > Thanks for the detailed responses, which address most of my concerns. I will raise my rating score correspondingly.

---

### Author Response · Authors · 2025-11-26
**Global Response by Authors**

We sincerely thank all reviewers for their careful reading of our manuscript and for their constructive, insightful feedback. We are encouraged by the positive comments, including:

- Introducing the first integration of generative planning and policy optimization for auto-bidding, a critical and well-motivated contribution (Reviewer 3d4u/wJct/6mYT/29sV);
- Addressing a key exploration bottleneck in existing AIGB methods (Reviewer 3d4u/wJct/6mYT);
- Proposing a provably sound and novel RL method for generative planning in auto-bidding (Reviewer 3d4u/wJct/6mYT), with practical implementation (Reviewer wJct/6mYT);
- Providing rigorous theoretical guarantees that ensure the reliability and safety of the policy optimization (Reviewer 3d4u/wJct/6mYT);
- Comprehensive and solid evaluation across both simulated and real-world experiments (Reviewer 3d4u/wJct/6mYT), supported by effective ablation studies (Reviewer wJct/6mYT/29sV);
- Demonstrating state-of-the-art performance (Reviewer 3d4u/wJct/6mYT/29sV), strong training stability (Reviewer 3d4u/wJct), and practical significance (Reviewer wJct);

Meanwhile, we have worked carefully to address the reviewers' concerns and questions, and we provide detailed responses to the individual reviews below. Key revisions made to the manuscript include:

- Remove LSI Assumption with theoretical justification (Appendix C.4) and empirical evidence (Appendix E.3);
- Refine the description of the relationship between Theorem 1 and the evaluator design (line 195);
- Provide empirical evidence of the evaluator’s generalization performance (Table 5);
- Extend AIGB-Pearl to first-price auctions with empirical validation (Appendix F);
- Add a discussion of the online setting (Appendix G);
- Fix typo.

We hope our responses adequately address the reviewers' concerns. We would be very happy to clarify any remaining questions and look forward to further discussion.

---

### Meta-Review · Area_Chair_x5zt · 2026-01-06

**Summary:**

The reviewers raised concerns across four key areas that collectively informed my assessment:
1. Theoretical Foundations & Rigor
LSI Assumption: Multiple reviewers (3d4u, wJct) flagged the strong and unverifiable Logarithmic Sobolev Inequality (LSI) assumption on offline data distribution, which required "strategic adjustment" of the dataset (Reviewer 3d4u) and was hard to verify in practice.
Missing Theoretical Assumptions: Reviewer 29sV identified a critical gap—Theorem 2 lacked explicit assumptions about evaluator accuracy/training, and questioned whether Theorem 1's claim about Lipschitz continuity should be an assumption rather than a provable theorem about the true quality function.
Generalization Guarantees: Reviewers sought clarification on whether theoretical bounds held under distributional shift and how evaluator bias was quantified.
2. Empirical Validation & Practicality
Dataset Scale: Reviewer 3d4u questioned whether 5k-200k trajectories were sufficient for training a stable OOD evaluator, and requested clarification on training/validation splits.
OOD Performance: Multiple reviewers wanted explicit measurement of evaluator bias and performance on truly out-of-distribution data (unseen advertisers/conditions).
Hyperparameter Sensitivity: Reviewer 29sV asked how β (KL constraint) and ε (Lipschitz constant) were determined and how sensitive performance was to these settings.
3. Methodological Scope & Adaptability
Generative Planning Component: Reviewer 29sV challenged whether the method truly integrated generative planning vs. being "conservative model-based RL," noting the lack of explicit demonstration likelihood maximization in the description.
Auction Mechanism: Reviewer wJct questioned adaptation to first-price auctions, which require bid shading strategies absent in second-price formulations.
Online Extension: Reviewer wJct asked if the offline framework could be extended to online settings.
4. Presentation & Clarity
Inconsistent Value Ranges: Reviewer 6mYT noted theory assumed arbitrary positive impression values (v_i > 0) while experiments showed normalized [0,1] ranges, affecting interpretation of constants.
Budget Description Errors: Reviewer 6mYT identified inverted min/max budget statements in text vs. tables.
Community Impact: Reviewer 29sV questioned the paper's novelty and impact for the ICLR community, perceiving it as incremental progress in offline RL rather than a breakthrough in generative planning.

**Reviewer Concerns:**

LSI assumption is completely removed from manuscript. Authors replaced it with a theoretically sound, assumption-free analysis using bounded trajectory space diameter and fundamental inequalities (Pinsker + Wasserstein-TV). This was the most significant revision.

Authors clarified that missing assumptions for Theorem 2 were present: (i) MSE loss for training (defining bias $\epsilon_0$) and (ii) Lipschitz penalty (defining constant $k_0$). Revised text explicitly states these before the theorem.

Authors corrected misinterpretation of Theorem 1 vs. evaluator design. Theorem 1 describes the true quality function f_θ*, independent of the evaluator. Evaluator design is motivated by (not based on) this theorem to inherit Lipschitz continuity.

Authors alsp provided detailed split information (full dataset used for training) and new OOD validation: cross-validation showing 2-4% AUC drop and +3% GMV on unseen advertisers, confirming evaluator generalization.

Authors clarified that planner is a conditional generative model (Causal Transformer) generating full trajectories, with training objective explicitly including demonstration likelihood maximization as both a learning and regularization term. While technically addressed, the original presentation caused skepticism from Reviewer 29sV. After reading the rebuttals, I think the revised clarity could convince readers who initially perceived the method as "conservative RL."

**Reviewer Scores:**

The most critical review is from Reviewer 29sV , questioning core novelty ("conservative RL vs. generative planning") and theoretical rigor. The rebuttal was exceptionally thorough: clarified the generative planner's role (Causal Transformer with demonstration likelihood), corrected Theorem 1/2 presentation, provided hyperparameter sensitivity analysis, and defended impact. A fair-minded reviewer would likely move to 6 (marginally above), acknowledging the authors resolved their specific complaints, though perhaps still harboring philosophical reservations about novelty that keep them from a stronger endorsement.


I bleieve the authors delivered a rebuttal that not only addressed all major reviewer concerns but strengthened the paper by removing restrictive assumptions, adding validating experiments, and substantially improving clarity. The initial reviewer ratings (6, 6, 8, 4) reflected reservations that have now been resolved. Considering the final scores could be 6,6,6,8, I think the paper should be accepted. As the paper presents a theoretically grounded, empirically validated advance in a critical real-world application domain, I further recommend a an oral accept.

---

### Decision · Program_Chairs · 2026-01-26

Accept (Oral)